


# Changes in flood damage with global warming in the east coast of Spain

Maria Cortès[1,2], Marco Turco[3], Philip Ward[4], Josep A. Sánchez-Espigares[5], Lorenzo Alfieri[6], and Maria Carmen Llasat[1,2]

[1]Department of Applied Physics, University of Barcelona, Barcelona, 08028, Spain
[2]Water Research Institute (IdRA), University of Barcelona, Barcelona, 08001, Spain
[3]Barcelona Supercomputing Center (BSC), Carrer de Jordi Girona 29-31, 08034 Barcelona, Spain
[4]Institute for Environmental Studies (IVM), Vrije Universiteit Amsterdam, De Boelelaan 1087, 1081HV Amsterdam, The Netherlands
[5]Department of Statistics and Operations Research, Technical University of Catalonia, Barcelona, Spain
[6]European Commission, Joint Research Centre (JRC), Directorate Space, Security and Migration, Ispra, Italy

**Correspondence:** Maria Cortès (mcortes@meteo.ub.edu)

**Abstract.** Flooding is one of the main natural hazard in the world and causes huge economic and human impacts. Assessing the flood damage in the Mediterranean region is of great importance, especially because its large vulnerability to climate change. Most past floods affecting the region were caused by intense precipitation events, thus the analysis of the links between precipitation and flood damage is crucial. The main objective of this paper is to estimate changes in the probability of damaging

flood events with a global warming of 1.5, 2 and 3 °C above preindustrial levels and taking into account different socioeconomic scenarios in two western Mediterranean regions, namely Catalonia and the Valencian Community. To do this, we analyse the relationship between heavy precipitation and flood damage estimates from insurance datasets in those two regions. We consider an ensemble of seven regional climate model simulations spanning the period 1976-2100 to evaluate precipitation changes and to drive a logistic model that links precipitation and flood damage estimates, and thus to derive statistics under present and

future climates. Furthermore, we incorporate population projections based on 5 different socioeconomic scenarios. The results show a general increase in the probability of a damaging event for most of the cases and in both regions of study, with larger increments when higher warming is considered. Moreover, this increase is higher when both climate and population change are included. When population is considered, all the periods and models show a clearly higher increase in the probability of damaging events, which is statistically significant for most of the cases.

Our findings highlight the need for limiting the global warming as much as possible, as well as the importance of including variables that consider change in both climate and socioeconomic conditions in the analysis of flood damage.



# 1   Introduction

In the Mediterranean region, intense precipitation events constitute a real danger to the population. Heavy precipitation, sometimes associated with strong winds, can cause floods with dire consequences for people and the environment (Fourrie et al., 2016) particularly during the autumn season. Most of these events are a consequence of short and local heavy rains in small

catchments, often near the coast in densely populated areas (Thiébault, 2018). The relief surrounding the Mediterranean Sea forces the convergence of low-level atmospheric flows and the uplift of warm wet air masses that drift from the Mediterranean Sea to the coasts, thereby creating active convection. In addition, population growth is particularly high along the Mediterranean coasts, leading to a rapid increase in urban settlements and populations exposed to flooding (Gaume et al., 2016).

In the last few years, much research has focused on the study of climate change in the Mediterranean region, an area that is

identified as highly vulnerable to climate change according to the Fifth Assessment Report of the Intergovernmental Panel on Climate Change (Pachauri et al., 2014). Some studies have found an increase in precipitation extremes with global warming projections in the Mediterranean region (Colmet-Daage et al., 2018; Drobinski et al., 2018), although a general decrease in the annual precipitation is projected (Jacob et al., 2014; Cramer et al., 2018; Sillmann et al., 2013; Rajczak and Schär, 2017). Cramer et al. (2018) state that future warming in the Mediterranean region is expected to exceed the global mean rates by 25

%, with peaks of 40 % in summer. As a result, flood risk associated with extreme precipitation events is expected to increase due to climate change in this area, but also due to non-climatic factors such as increasingly sealed surfaces in urban areas and ill-conceived storm-water management systems (Cramer et al., 2018).

A large number of floods affecting the western Mediterranean region are surface water floods (Llasat et al., 2014; Cortès et al., 2018). This type of flood can be regarded as coming under the most general definition of rainfall-related floods (Bernet

et al., 2017; Cortès et al., 2018). For this reason, precipitation is the main hazard driver of the damage caused by these events.

Nevertheless, flood disasters are the result of both societal and climatological factors, hence several other drivers other than climate must be considered for the assessment of flood-damage trends (Barredo, 2009). Bouwer (2011) analyses 22 disaster loss studies around the globe, showing that economic losses from various weather-related natural hazards have increased. However, most of these studies have not found a trend in disaster losses after normalization for changes in population and wealth, pointing

towards increasing concentrations and values of assets as the principle cause of the increasing damages and losses from natural disasters. In Spain, the findings of Barredo et al. (2012) align with these results; they find no significant trend in adjusted insured flood losses between 1971 and 2008. These studies show the need to include exposure and vulnerability changes in future risk projections, which clearly contribute substantially to changing risks.

Insurance data may provide a good proxy for describing flood damage (Barredo et al., 2012), and several studies note its

potential for describing economic damages caused by surface water and urban floods (Spekkers et al., 2013; Torgersen et al., 2015; Cortès et al., 2018). However, there are still few studies that relate flooding with insurance data, since these companies are reluctant to provide their databases and some of them are not even available due to confidentiality restrictions (André et al., 2013; Leal et al., 2019). Nevertheless, in the Mediterranean region, 20 years of flood-related insurance damage claims are available from the Spanish public reinsurer, the Insurance Compensation Consortium (Consorcio de Compensación de





Seguros, CCS), for the Spanish regions of Catalonia and the Valencian Community. CCS is a public institution that compensates homeowners for damage caused by floods, playing a role similar to that of a reinsurance company (Barredo et al., 2012). Therefore, in this study, insurance data is used as a proxy for flood damage.

The main objective of this paper is to estimate changes in the probability of damaging flood events with global warming of 1.5, 2 and 3 °C above the preindustrial levels, taking into account different socioeconomic scenarios. To do this, we analyse the relationship between heavy precipitation and flood damage estimates from insurance datasets in two western Mediterranean regions, namely the Valencian Community and Catalonia regions of Spain. We use an ensemble of seven regional climate model simulations and 5 different socioeconomic scenarios to study future changes in these relationships.

This article is organised as follows. After an Introduction, the Methods section describes the study region, the data used and the methodology describing the model developed for the present climate as well as the treatment applied for the future precipitation and population projections. Then, the Results and discussion section presents the statistical models developed, the analysis of the future data, and the probability of flood damage with global warming, comparing our results with those obtained in other studies. Finally, a section describing the limitations of the study and possible future research is presented followed by the Conclusions, which summarises the main findings of this study.

## 2  Methods

Figure 1 describes the overall methodology followed in this study and the data used. After selecting the flood events that have affected the region of study, we collected information on damage (insurance payments), hazard (precipitation) and exposure (population) for each basin affected by these events (Panel 1). These data have been used to develop Generalized Linear Mixed Models in order to assess the probability of damaging events for the present climate (Panel 2). Precipitation and population data from future projections were corrected (further information in Section 2.4) and aggregated at the river-basin scale (Panel 3). Finally, we assessed changes in the probability of a damaging event with respect to the reference period (1976-2005) with global warming of 1.5, 2 and 3 °C above the preindustrial levels (Panel 4) using the relationships found in the present climate (Panel 2) and the data from future projections (Panel 3). In Section 2.2, the data and their sources are explained in detail.





**Figure 1.** Scheme of the overall methodology followed in the study

## 2.1   Region of study

The domain of this study is the eastern coast of Spain, and consists of Catalonia and the Valencian Community (Figure 2). The Mediterranean presents a complex orography and particular location - at the transition area between extra-tropical and subtropical influence (Giorgi and Lionello, 2008) - that leads to a great variety of climates with both Atlantic and Mediterranean

5   influences. Thus, precipitation is characterised by a complex spatial pattern, with a strong seasonal cycle and large interannual (Trigo and Palutikof, 2001) and spatial variability (Rodriguez-Puebla et al., 1998; Romero et al., 1998; Martin-Vide, 2004; Rodrigo and Trigo, 2007; Quintana-Seguí et al., 2016, 2017). Due to this strong variability, this region represents a challenging area for downscaling of precipitation studies (see e.g. Turco et al., 2011). Catalonia, situated in the north-east of the Iberian Peninsula has a surface of 32,108 km$^2$ and a population of more than 7.5 million people (IDESCAT, 2018). The coastal zone is



a very vulnerable area since it is highly densely populated, with municipalities such as Barcelona that have a population density around 16,000 inhabitants $km^{-2}$. Catalonia is characterised by three mountain ranges (Figure 2, Nº 1): the Pyrenees in the north (maximum altitude above 3,000 m a.s.l.); a range parallel to the Mediterranean coast (SW-NE) named the Catalan Pre-Coastal Range (maximum altitude around 1,800 m a.s.l.); and the Catalan Coastal Range (maximum altitude around 600 m a.s.l.). This

5    marked orography is one of the key reasons for producing floods, both from a hydrological point of view, since presents small torrential catchments, and from a meteorological point of view, as the orography forces wet air to rise from the Mediterranean (Llasat et al., 2016). The presence of the Pyrenees can also lead to remarkable effects in the mesoscale pressure distribution, giving place to processes such as convergence lines and orographic dipoles. For example, the flood event of October 1987, when more than 400 mm of rainfall were recorded in 24 h near Barcelona, was favoured by a mesohigh created at the south

10   of the Pyrenees (Ramis et al., 1994). For the 1996-2015 period, a total of 166 flood events were recorded in Catalonia, 13 of them caused catastrophic impacts, 87 extraordinary impacts and 66 ordinary impacts (Cortès et al., 2018), following the methodology explained in Llasat et al. (2016) and Cortès et al. (2017).

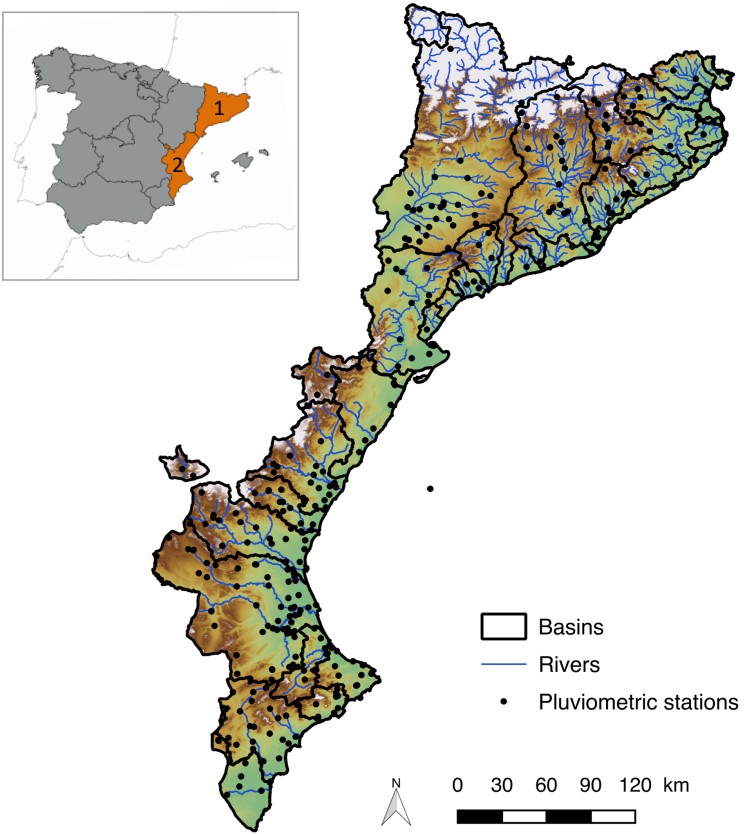

**Figure 2.** Map of both regions of study (1: Catalonia; 2: Valencian Community), showing the aggregated basins (black lines), the main rivers (blue lines) and the pluviometric stations used (black points)




The second region of the study is the Valencian Community (Figure 2, Nº 2), with a population around 5 million people and a total surface of 23,255 km² (INE, 2018). Similar to the Catalonia region, the coastal plains are crossed by torrential non-permanent streams, around which intense urbanization has taken place. In the South of the region, the mountains almost reach the coastline (maximum altitude around 1,560 m a.s.l.), favouring heavy precipitations that can exceed 700 mm in 24

h. This was the case in November 1987, when more than 800 mm of rainfall were recorded in 24 hours in Oliva (Valencia) (Ramis et al., 2013). The Valencian Community recorded a total of 69 flood events for the 1996-2015 period, 11 of which were catastrophic, 26 extraordinary and 32 ordinary (Cortès et al., 2018).

## 2.2   Data

We used three different databases to select the flood events that affected the regions of study (Catalonia and Valencian Commu-

nity): INUNGAMA (Barnolas and Llasat, 2007; Llasat et al., 2016), PRESSGAMA (Llasat et al., 2009) and FLOODHYMEX (Llasat et al., 2013). INUNGAMA is a database of the GAMA (*Grup d'Anàlisi de situacions Meteorològiques Adverses*) group, which reports the flood events that have occurred in Catalonia from 1900 to 2015 on a municipal, county, and basin level. Information contained in the database includes hydrometeorological data, the impacts caused, and the affected areas for each of the events. The major part of this information is provided by PRESSGAMA, a database formed from press data, and other

data sources such as official reports. This database, which includes more than 15,000 news items, systematically collects information on natural risks and climate change in the newspaper *La Vanguardia* from 1981 to 2015. Finally, for the flood events recorded in the Valencian Community, we used the FLOODHYMEX database, which contains both hydrometeorological and impacts information produced by flood events that have affected different Mediterranean regions for the period 1980-2015.

Population data for Catalonia were obtained from the Statistical Institute of Catalonia (*Institut d'Estadística de Catalunya*,

IDESCAT, 2018). In the case of the Valencian Community, the data were provided by the Spanish National Statistics Institute (*Instituto Nacional de Estadística*, INE, 2018). The population data corresponds to the year when the flood event took place.

We have used daily precipitation data provided by the Spanish State Meteorological Agency (*Agencia Estatal de Meteorología*, AEMET), which has an extensive network of automatic meteorological stations spread throughout the entire region of study collecting the accumulated daily precipitation between 7 UTC and 7 UTC of the following day. To ensure temporal

homogeneity, we have only considered the stations with more than 90 % valid data over the 1996-2015 period (Figure 2).

Flood damage data were obtained from the insurance compensations due to floods paid by the Spanish Insurance Compensation Consortium (*Consorcio de Compensación de Seguros*, CCS). The CCS compensates for damage caused to people and property by floods and other adverse weather events covered by an insurance policy. These data are provided at postcode level and with a daily temporal resolution. The CCS database includes around of 58,000 records of claims paid for floods in Catalo-

nia and more than 100,000 in the Valencian Community for the 1996-2015 period (no previous information is available with this level of detail). To compare these data with other variables, we aggregated the damage in each postcode to the municipality level and then calculated the total amount per each flood event and basin affected. This process was carried out taking into account the claims made for the days on which the event occurred (according to INUNGAMA and FLOODHYMEX databases), and the following seven days. We used this 7-day window as this is the period of time that the CCS allows insurance claims to





be made. When the time difference between the two events is less than 7 days, damages are associated with the first event, if the date of the claim was before the first day of the second event. Finally, damage data were adjusted to 2015 values, following the methodology defined by the Spanish National Statistics Institute (INE, 2018). This consists of using the exchange rate in the consumer price index (CPI) between the two years to adjust the values shown in Euros.

Daily precipitation data for 7 climate projections were obtained from an ensemble of simulations developed within the EURO-CORDEX project (Jacob et al., 2014), which covers the whole of Europe with a spatial resolution of 0.11 degrees in latitude and longitude (around 12 km). Tramblay et al. (2013) observed an improvement in the representation of precipitation and its extremes at this resolution compared with previous simulations at 25 and 50 km resolution. Overall, the seven climate projections are combinations of three different General Circulation Models (GCM), which were then downscaled with four

RCMs as shown in Table 1. The models chosen are the ones that had the necessary variables at the moment of the design of our study, and they have been extensively validated in previous studies (see, e.g., Jacob et al., 2014; Alfieri et al., 2015b, 2018; Jerez et al., 2018) and used to study climate change impacts (Jerez et al., 2015). The Representative Concentration Pathway 8.5 (RCP8.5) from the Intergovernmental Panel on Climate Change (Pachauri et al., 2014) is the scenario used from 2006 to 2100 for the future projections. For the historical simulations (or reference period) the 1976-2005 period has been used.

The five Shared Socioeconomic Pathways (SSP) have been applied for the estimation of the population change in the future. The SSP projections (O'Neill et al., 2014) have been developed by the research community to facilitate integrated assessments of climate impacts, vulnerabilities, adaptation and mitigation. These socioeconomic pathways include projections for population, urbanization and Gross Domestic Product at global and national scales. In order to obtain this information at the spatial resolution required for the study (basin level), gridded data were used from the 2UP (Towards an Urban Preview) model,

developed by the Netherlands Environmental Assessment Agency. The core of the 2UP model and its primary objective is to disaggregate the scenario-based projected national-level urban population to 30 arcseconds (approximately 1 km near the equator) and simulate urban expansion for 194 countries and territories. This high spatial resolution grid has been used to calculate the future total population in each basin of the regions of study and for the different SSPs.

## 2.3   Modelling damage probabilities for the present climate

### 25  2.3.1   Generalized Linear Mixed Model

The aim of this section is to develop a model that is able to gauge the probability of large damaging events occurring given a certain precipitation amount and taking into account other variables related with the exposure of the territory. That is, our aim is not to estimate the precise amount of the monetary compensation, but to estimate when a 'large' damaging event will occur given a certain precipitation amount. Since there is not a standard definition of a large damaging event, we tested several cases,

corresponding to insurance compensations exceeding the 10th, 20th, 30th, 40th, 50th, 60th, 70th, 80th and 90th percentile of the total sample. The minimum geographical unit of the study is the river-basin-scale and we only consider the flood cases that recorded a mean precipitation in the basin higher than 40 mm in 24 h. Barbería et al. (2014) suggest that with a threshold of 40 mm 24 h[-1], social impacts are expected for rain events in urban areas of Catalonia, and Cortès et al. (2018) found a strong





relationship between insurance data and precipitation when using this threshold for assessing the flood damage in the whole Catalan region. Population data was also taken into account, to model flood exposure in the region. Therefore, we sampled the response variable (i.e. the compensation series) and both explanatory variables (the mean 24 h precipitation recorded for each basin and the total population of the basin), and pooled them into one sample for each entire region (Catalonia and Valencian

Community) to correlate them. For each event there can be more than one set of values, depending on the number of affected catchments. From now on we will use the expression "flood case" for each set of values corresponding to a basin affected by a flood event. This area is large enough to have a fairly large sample size for the analysis, yet small enough so that the causes of flood damage are related to the same weather pattern.

In order to accomplish this objective, a Generalized Linear Mixed Model (or GLMM) has been applied for both regions and

all the percentiles of damage. The GLMMs are a powerful class of statistical models that combine the characteristics of linear mixed models (models that include both fixed and random predictor variables) and generalized linear models (which handle non-normal data by using link functions and exponential family distributions [e.g. Poisson or binomial]). Thus, GLMMs are the best tool for analysing non-normal data that involve random effects (Bolker et al., 2009). That is the case with the data used in this research, which is of binary type (event or non event) and there are random effects related to space and time: each of the

basins can be affected by different flood events for the whole time period, and moreover each event can affect different basins at the same time. This implies high correlations within the observations, not guaranteeing the independence requirement of the Generalized Linear Models (GLM).

The generalized linear mixed model follows the equation 1:

$$\log(\frac{\pi}{1-\pi}) = \beta_0 + \beta_1 P + \beta_2 R + b_i + b_j \tag{1}$$

Where $\pi$ is the response variable (the probability to exceed a specific threshold of damage), $P$ and $R$ are the predictors (precipitation recorded in 24 h and total population of the basin in our case), and $b_i$ and $b_j$ are the random effects related to the basins and the events. The value of the $\beta$ coefficient is determined using Generalized Linear Models (GLMs). The Wald $\chi^2$ statistic is used to assess the statistical significance of individual regression coefficients (Harrell Jr, 2015).

### 2.3.2 Validation method

We plotted the relative operating characteristic (ROC) diagram, a commonly used logistic prediction diagnostic, showing the hit rate (i.e. the relative number of times a forecasted event actually occurred) against the false alarm rate (i.e. the relative number of times an event had been forecasted but did not actually happen) for different potential decision thresholds (Mason and Graham, 2002). Thus, for each insurance compensation percentile, we first calculated the forecast probabilities for that event, and then grouped the probability forecasts into batches (here 100 with a width of 0.01) to count the observed

occurrences/non-occurrences. That is, we converted the observed and forecasted series, expressed as continuous amounts, into exceedance categories (yes/no statements indicating whether the data equal or exceed a selected probability). We then plotted the resulting elements on a standard contingency table (see Table 2). The ROC diagram shows the hit rate (H) against the false



alarm rate (F). These indices are defined as

$$H = \frac{a}{a+c}; 0 \leq H \leq 1 \tag{2}$$

$$F = \frac{b}{b+d}; 0 \leq F \leq 1 \tag{3}$$

## 2.4 Future probability of flood damage

### 2.4.1 Precipitation extremes in future climate change scenarios

In this study, we analysed changes in daily precipitation data from the EURO-CORDEX simulations assuming global warming scenarios of 1.5, 2 and 3 °C above preindustrial levels (1881-1910) (Vautard et al., 2014). For each simulation, the year when a 20-year running mean of global average temperature exceeds 1.5, 2, and 3 °C in the RCP8.5 dataset is identified and then a 30-year time window applied for each warming period and model. Therefore, for each simulation we have 3 periods of 30 years, described in Table 1, as well as the reference period 1976-2005. This method is based on the guidelines of the HELIX project (Betts et al., 2018).

To select the precipitation data to be used in the analysis, we first aggregate all the time-series to the river-basin level, calculating the mean of the grid cells belonging to each of the basin polygons. Then, we applied a bias correction to smooth the differences between observations and simulations. To this end, we selected the common period 1996-2005. The data from the simulations comes from the historical data series of each model, while the data from the observations is the daily precipitation data from the AEMET weather stations with an effectiveness higher than 90 %. Both datasets are aggregated to the river-basin-scale, using the mean value. Once the two pairs of comparable series are available for each model, the differences between them are estimated by applying a quantile mapping bias correction method, specifically with a non-parametric quantile mapping using the robust empirical quantiles method (Gudmundsson et al., 2012). This method estimates the values of the quantile-quantile relation of observed and modelled time series for regularly spaced quantiles using local linear least square regression. In order to check the suitability of the method chosen, the differences between the observations and simulations for the period 1996-2005 have been compared with and without the correction applied. This was done for the cases that recorded a mean precipitation in the basin higher than 40 mm, since is the same sample used in the development of the model for the present climate.

Figure 3 shows an example for the Valencian Community, with a comparison between the simulations and observed precipitation for the common period 1996-2005 for the flood cases that exceeded the 40 mm 24 h$^{-1}$ precipitation threshold without (top panel) and with (bottom) the correction applied. The homogenization achieved with the bias correction process is clearly reflected in the bottom panel of the Figure 3.

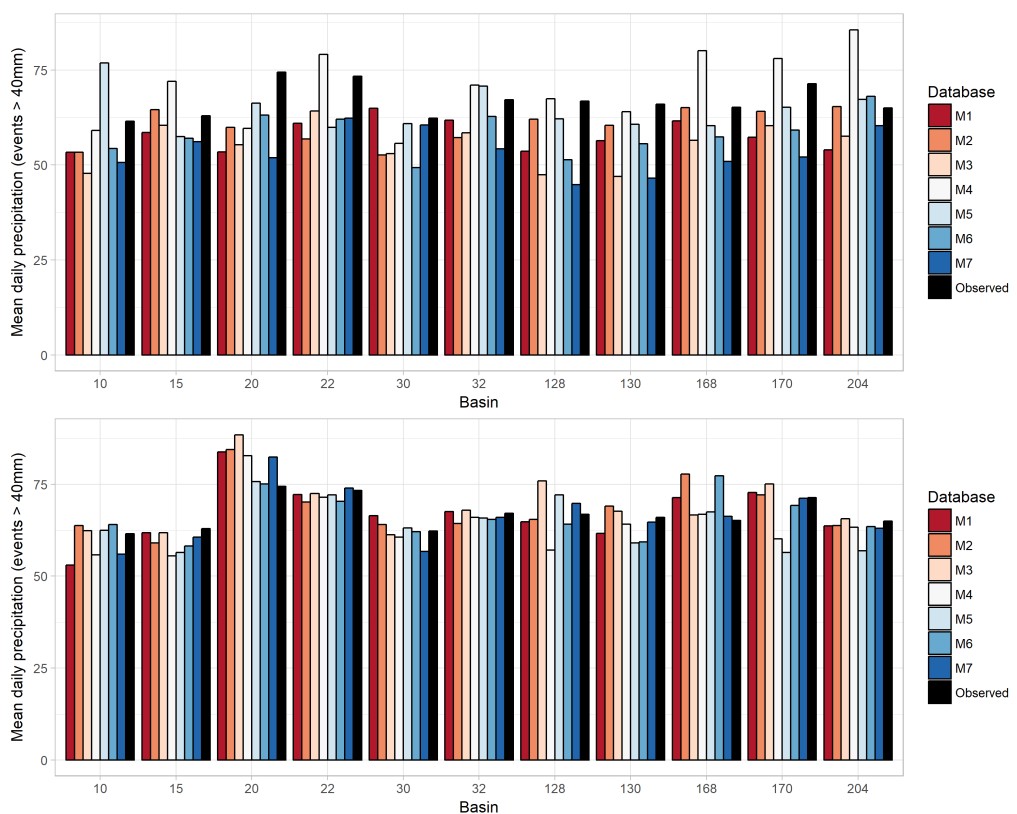

**Figure 3.** Comparison of the mean precipitation value (cases with average daily precipitation exceeding 40 mm) for each basin of the Valencian Community between the observations and the different models with (bottom) and without (top) correction for the common period 1996-2005. M1 to M7 refer to the 7 EURO-CORDEX simulations used (Table 1). The numbers on the x-axis indicate the code of each basin (see Table A2, Appendix section)

The same bias correction method has been applied for the three future periods (1.5 °C, 2 °C and 3 °C), using the specific correction of each basin and model. Finally, only the cases that recorded a mean precipitation in the basin higher than 40 mm 24 h$^{-1}$ were selected, in order to use the same data as those that were introduced to build the damage model for the present climate (Section 2.3).

5 **2.4.2 Population projections**

The Shared Socioeconomic Pathways (SSPs), together with the 2UP model (van Huijstee et al., 2018), have been used to estimate the total population of each of the basins of the study for the future projections. The main objective of the 2UP model is to disaggregate scenario-based projected national-level urban population to 30 arcseconds data and simulate urban expansion for 194 countries and territories. In order to incorporate this data in the damage model, three main processes have been applied: 10 (i) temporal downscaling from 10 years resolution to yearly; (ii) river-basin-level aggregation; and (iii) bias correction.





For the temporal downscaling, simple linear regression has been applied in order to obtain yearly population data for each of the SSPs. The 2UP model estimates the population every 10 years, from 2010 to 2080. When climate projections for the + 3 °C scenario exceed the year 2080, the same population change coefficient for the last ten years period (2070-2080) has been applied.

After obtaining the yearly population data for all SSPs, the data were aggregated to the river-basin-level by summation. Therefore, each basin will have 5 different yearly population series (corresponding to each SSP) from 2010 to 2084 (the last year of the precipitation simulations).

Finally, a bias correction process is also required for the population data. The year 2010 was chosen for carrying out the comparison between the population data from the observations and those from the 2UP model. Figures A1 and A2 from the Appendix section show the 2010 population aggregated at basin and region level for Catalonia and the Valencian Community, respectively. As can be observed in both cases, differences between the projections are negligible, though there exists a significant disagreement with the observations. For this reason we used a scaling bias correction method to correct the population projection data. This means that the ratio for each basin and SSP between the observed and simulated population for the 2010 year was applied to the future population projections.

Therefore, after these processes, each basin has a yearly population series corrected from 2010 to 2084 for the 5 SSP.

## 3 Results and discussion

### 3.1 Present climate

#### 3.1.1 Catalonia

We applied the Generalized Linear Mixed Model to all the flood events that affected Catalonia basins within 1996-2015, which resulted in a total of 596 flood cases, 177 of which recorded an average precipitation in the basin higher than 40 mm.

The regression coefficients for all the percentiles of damage obtained by applying the GLMM are given in Table 3. In all cases both of the predictors have positive values, showing that the probability of a damaging event increases not only with precipitation recorded in 24 h but also with the population of the basin.

The formula considering the 70th percentile of damage is shown as an example in Equation 4:

$$\log(\frac{\pi}{1-\pi}) = -20.92 + 2.98\log(P) + 0.57\log(R) + b_i + b_j \qquad (4)$$

Where $\pi$ is the probability of exceeding the 70th percentile of damage (in this case), $P$ the mean precipitation accumulated in the basin in 24 h, $R$ the total population of the basin and $b_i$ and $b_j$ are the random effects related to the basin and the flood event.

Both precipitation and population are statistically significant, meaning that they are useful variables to explain the probability of exceeding the 70th percentile of damage (0.38 M Euro). This can be observed in the examples in Figure 4, which show the


effect of each explanatory variable in the case of probability exceeding the 70th percentile of damage. The solid line indicates the best estimate while the shaded blue bands show the 95 % confidence interval. For both variables, this probability increases rapidly. However, in the case of population a break point can be observed around 0.5 million people, above which the rate of increase is much lower.

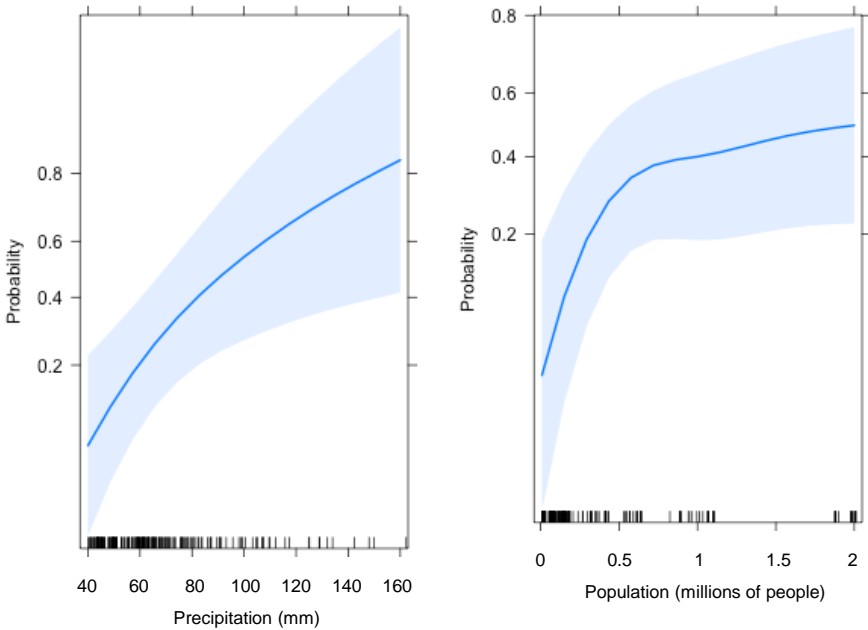

**Figure 4.** Effect of the explanatory variables (left: mean precipitation recorded in 24 h; right: total population of the basin) on the probability of exceeding the 70th percentile of damage in Catalonia. The solid line indicates the best estimate while the shaded blue bands indicate the 95 % confidence interval. The black marks at the bottom of each graph represent the values of the independent variable for each flood case

5      The area under the ROC curve (RA) has been used to validate the model, since it is a useful measure for summarising the skill of a model. RA ranges from 0 for a forecast with no hits and only false alarms, to 1 for a perfect forecast. Models with an RA above 0.5 have more skill than random forecasts. Figure B1 (Appendix section) shows the ROC diagram for the example shown in Equation 4. The RA value (0.95) indicates that our model has good fit to simulate the probability of a damaging event (defined as the 70th percentile of damage in this case). In this example, if the user wants to maximise the difference between

10    the hit rate (0.94) and the false alarm rate (0.17), a probability value of 0.22 is needed (which we called best threshold). The RA indicator for the different damage percentiles used is presented in Table 3, with values close to 1 for all the cases.





### 3.1.2 Valencian Community

The same analysis has been carried out for the Valencian Community region, which was affected by 69 flood events between 1996 and 2015, resulting in 171 flood cases (72 if we take into account only the cases where the mean precipitation accumulated in 24h in the basin exceeded the threshold of 40 mm 24 h$^{-1}$).

5     Table 4 shows the regression coefficients for all the damage percentiles used. As in the case of Catalonia (Section 3.1.1), both mean precipitation accumulated in 24h and total population of the basin are statistically significant predictors and have positive regression coefficients for all the cases.

As an example, the model follows Equation 5 when the damaging event is defined using the 70th percentile of damage:

$$\log(\frac{\pi}{1-\pi}) = -40.96 + 2.33 \log(P) + 2.32 \log(R) + b_i + b_j \qquad (5)$$

10    Figure 5 corroborates these results, showing the effect of precipitation (left graph) and population (right graph) on the probability of exceeding the 70th percentile of damage. In both cases this relationship is described by a concave curve, indicating that the increase in the probability per unit of increase of the predictor is larger when considering lower predictor values.

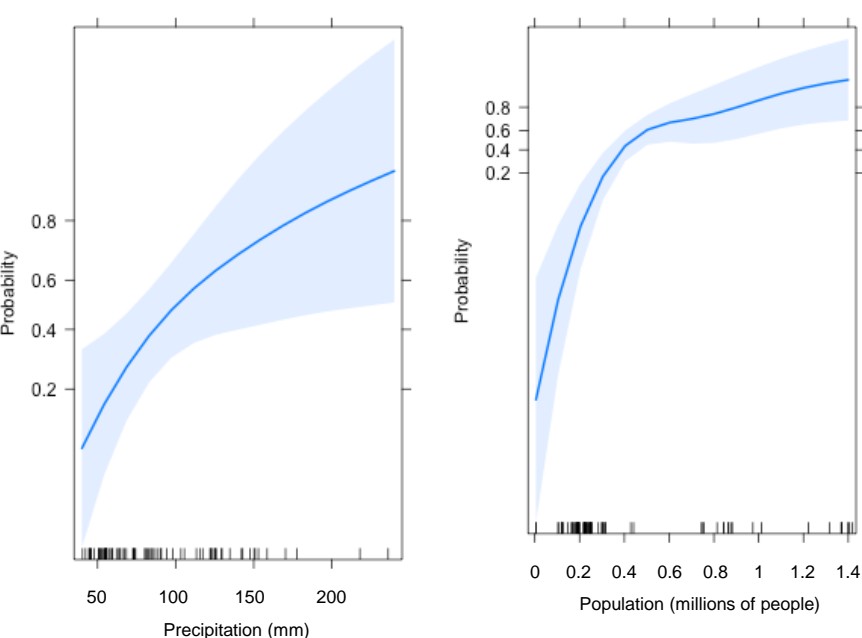

**Figure 5.** Effect of the explanatory variables (left: mean precipitation recorded in 24 h; right: total population of the basin) on the probability of exceeding the 70th percentile of damage in the Valencian Community. The solid line indicates the best estimate while the shaded blue bands indicate the 95 % confidence interval. The black marks at the bottom of each graph represent the values of the independent variable for each flood case





The shaded blue bands of Figure 5 show the confidence level of the prediction at 95 %. In this case, these bands are narrower than in the Catalonia case (see Figure 4), showing that the model estimation is more precise for the Valencian Community region. Figure B2 (Appendix section) displays the ROC diagram for the example of Equation 5, demonstrating that the model has a significant goodness of fit with a RA value of 0.9. As shown in Table 4, the model has a good performance for all the damage percentiles, with RA values close to 1 in all cases.

In this section we have shown that precipitation is a key factor in explaining the damage caused by flood events in regions in which water surface floods are the main type of flood, as is the case in the Mediterranean region of study. Studies like Zhou et al. (2013) and Torgersen et al. (2015) also apply regression models to observe the relationship between different precipitation observation measures and the compensations paid by insurance companies. In both studies, a good correlation is observed between these two variables in the case of surface water floods, and it is shown that it is feasible to build models that explain the costs of damage as a simple function of precipitation (Zhou et al., 2013). Moreover, the incorporation of population data in the model, with statistically significant and positive relationships with damages, has shown the importance of considering the exposure of the territory in flood risk analysis. Saint Martin et al. (2016) has also shown the benefits of including indicators that explain the exposure to assess the risk of flood-related damage in real time in French Mediterranean basins.

Our findings also align with the results of previous studies (Spekkers et al., 2013; Zhou et al., 2013; Wobus et al., 2014; Torgersen et al., 2015) and further indicate that insurance databases are a promising source for flood damage assessment at local (Garrote et al., 2016; Bihan et al., 2017; Zischg et al., 2018; Zhou et al., 2013) and regional scale (Barredo et al., 2012; Kim et al., 2012; Wobus et al., 2014; Zhou et al., 2017). However, using insurance data as a unique source for defining flood damage leads to only considering the direct and tangible damages. Indirect damages are more difficult to calculate and few studies take them into account (Elmer et al., 2010), especially at the regional scale (Przyluski and Hallegatte, 2011). In spite of this, some exhaustive studies can be found in the literature that incorporate indirect damage indicators such as, for example, the loss of working hours (Petrucci and Llasat, 2013) or the number of fire service operations done in flooded properties (Papagiannaki et al., 2015).

## 3.2 Future model

The results presented in this section are divided into three parts. In the first and second part, the data obtained from the precipitation projections (Section 3.2.1) and population scenarios (Section 3.2.2) are analysed. Then, the last part (Section 3.2.3) shows the results obtained by applying the models developed in Section 3.1 using the data from the projections. Therefore, in this last part, the probability of a damaging event is estimated for a global warming of 1.5, 2 and 3 °C and taking into account different socio-economic scenarios.





### 3.2.1 Extreme precipitation projections

Figure 6 shows the ensemble mean of the relative changes in the total annual precipitation (top) and the number of days with daily precipitation higher than 40 mm (bottom) for the entire Iberian Peninsula and for a global warming of 1.5, 2 and 3 °C above preindustrial levels.

Results show a general decrease of the total annual precipitation in most of the Iberian Peninsula, especially in the central and southern part. The decrease becomes larger with higher levels of global warming. This pattern is also observable in other studies of the Mediterranean region (Jacob et al., 2014; Cramer et al., 2018; Sillmann et al., 2013; Rajczak and Schär, 2017; Turco et al., 2017). For example, Kjellström et al. (2018) found an increase in mean precipitation in the north of Europe, but a reduction in the south, especially during summer months. In addition, they found a likely decrease in the mean precipitation

with global warming in mountain regions, such as the Pyrenees and Cantabrian Mountains. In terms of extremes, our results show projected increases in the number of days with daily precipitation exceeding 40 mm in the centre, north and north-west of the region (bottom panel Figure 6). This is also in line with other studies on projections of extreme precipitation in the region (Drobinski et al., 2018; Colmet-Daage et al., 2018; Tramblay and Somot, 2018; Cramer et al., 2018), which show a likely increase in the extreme precipitation events in Europe and the Mediterranean area with increasing global temperature, mostly

during the winter months (Kjellström et al., 2018; Vautard et al., 2014; Rajczak and Schär, 2017; Jacob et al., 2014).

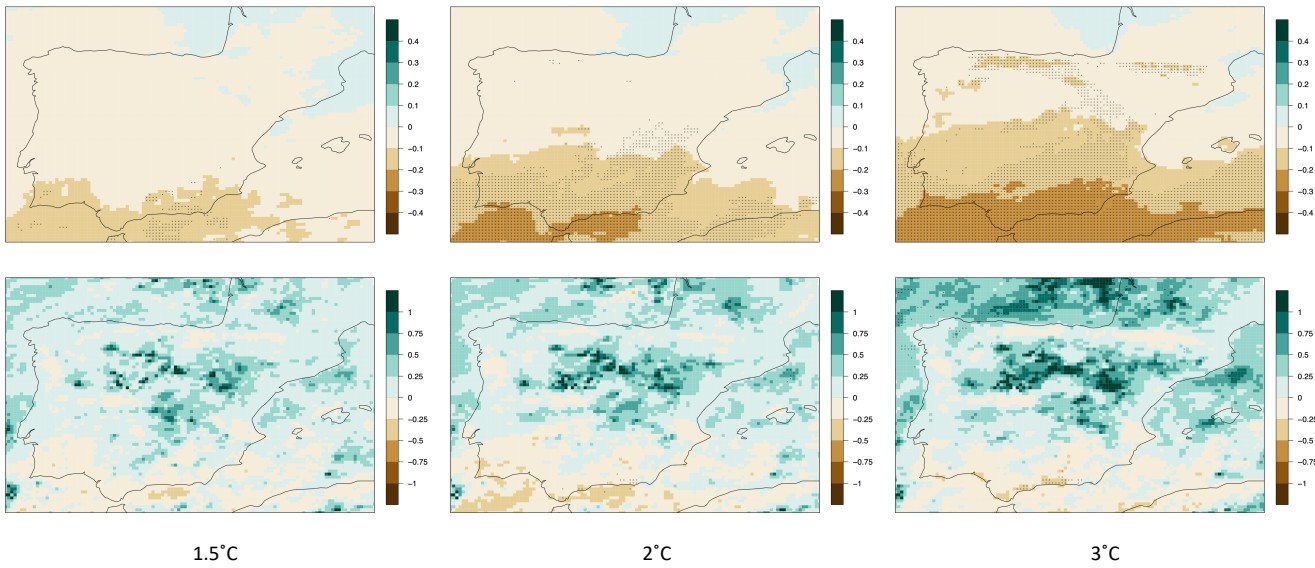

**Figure 6.** Ensemble mean of the relative changes of total annual precipitation (top) and number of days with daily precipitation > 40 mm (bottom) for 1.5, 2 and 3 °C global warming (columns) at annual scale. Dotted areas indicate where at least 50 % of the simulations show a statistically significant change and more than 66 % agree on the direction of the change. Coloured areas (without dots) indicate that changes are small compared to natural variations, and white regions (if any) indicate that no agreement between the simulations is found (similar to Tebaldi et al., 2011)

Figure 7 shows the change (in %) of the mean 24 h precipitation, averaged across all basins, in the future simulations compared to the reference period (1976-2005), when taking into account the cases that overpassed the 40 mm 24 h$^{-1}$ precipitation threshold for the different models and warming periods (global warming at 1.5 °C, 2 °C and 3 °C). In the case of Catalonia (left panel), most of the models and warming periods show an increase of the mean daily precipitation with the exception of

5 Model 3 (EC-EARTH-RCA4), which projects a decrease for the 2 °C global warming level. Most of the models (4 out of 7) indicate greater precipitation values when the highest warming level is taken into account (3 °C), agreeing with several studies that point to an increase of precipitation extremes with global warming (Drobinski et al., 2018; Colmet-Daage et al., 2018; Tramblay and Somot, 2018; Cramer et al., 2018).

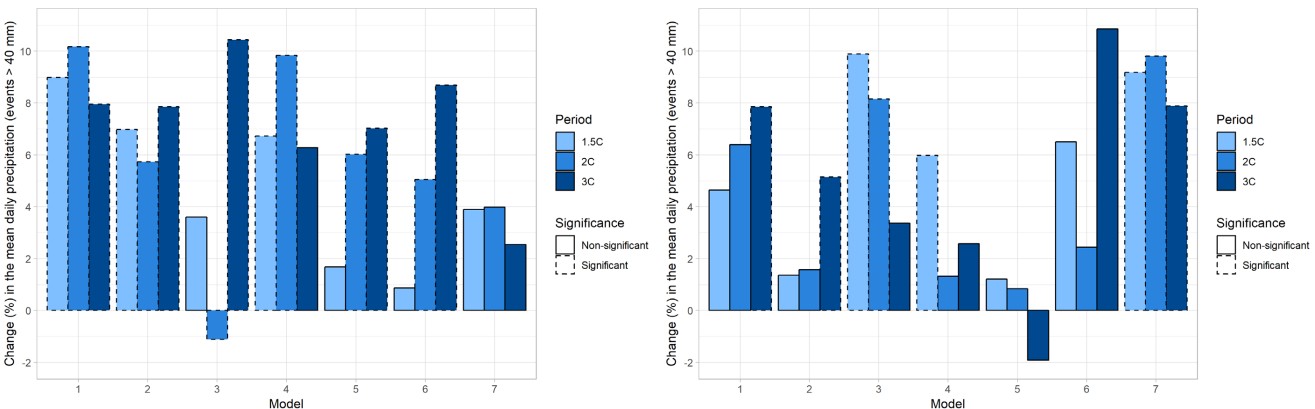

**Figure 7.** Change in the mean daily precipitation compared to the reference period (1976-2005) for cases exceeding the 40 mm 24 h$^{-1}$ threshold for the 7 models and the 3 levels of warming in Catalonia (left) and the Valencian Community (right). Dashed hatchings around bars indicate significant changes (p value < 0.05; Mann-Whitney Test) with respect to the reference period

In the case of the Valencian Community region, the change in the mean daily precipitation for the future periods is lower

(Figure 7, right panel). However, apart from one case which is non-significant (3 °C, Model 5: MPI-ESM-LR-CCLM4-8-17), all the models and periods show an increase of the mean precipitation of the basin when the events with daily precipitation exceeding the 40 mm are taken into account. In this case, there is no clear pattern of larger increases in extreme precipitation with higher warming levels.

The greater positive changes in precipitation values in the case of Catalonia was expected if we take into account the results

showed in Figure 6. The east of the Iberian Peninsula (where the Valencian Community is placed) presents a greater decrease in both annual precipitation and in the number of days with daily precipitation higher than 40 mm in comparison to the northeast part (Catalonia). Nevertheless, from these results we can conclude that heavy daily precipitation is likely to increase with global warming in both regions of study.





### 3.2.2 Population projections

The Shared Socioeconomic Pathways (SSPs) describe plausible alternative changes in aspects of society such as demographic, economic, technological, social, governance and environmental factors (O'Neill et al., 2017). The SSPs are based on five narratives describing alternative socio-economic developments, including sustainable development (SSP1), middle-of-the-road

development (SSP2), regional rivalry (SSP3), inequality (SSP4) and fossil-fueled development (SSP5) (Riahi et al., 2017).

Figure 8 shows the projections in the population in each entire region of study (left: Catalonia, right: Valencian Community) after applying the data treatment processes explained in Section 2.4.2.

For both regions, the largest increase in population is in SSP5, with population growth around 50 % for Catalonia and 62 % for the Valencian Community by the end of the 21st century with respect to 2010. SSP5 is defined by the push for economic

and social development coupled with the exploitation of abundant fossil fuel resources and the adoption of resource and energy intensive lifestyles around the world (O'Neill et al., 2017). Though fertility declines rapidly in developing countries, fertility levels in high income countries are relatively high. For this reason, despite being the scenario that projects the second lowest population by the end of the 21st century in terms of global population (Samir and Lutz, 2017), in our region of study it projects the highest increase (high-income economy).

On the other hand, SPP3 depicts the largest decrease in the total population in Catalonia and the Valencian Community, with a reduction of 25 % and 30 %, respectively. In this scenario a low population growth is expected in industrialized countries, while a high increase can be found in developing countries (O'Neill et al., 2017). This explains the difference between the results found in studies such as Samir and Lutz (2017) in which the highest global population expected growth is found for this scenario.

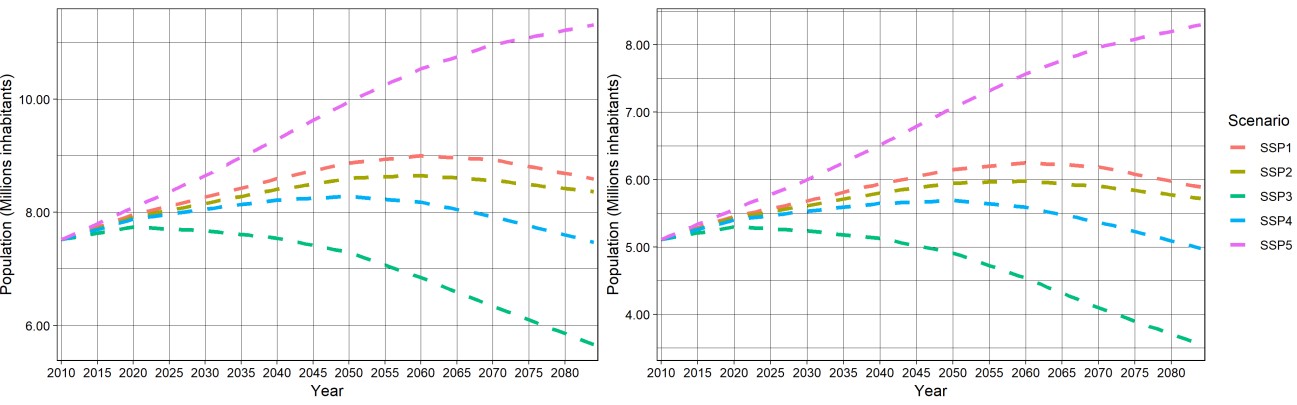

**Figure 8.** Future population projections for the different SSP's in Catalonia (left) and the Valencian Community (right)

SSP5 assumes a society that keeps relying heavily on fossil fuels. This implies that we may expect high emissions, and therefore the combination with RCP8.5 (the only RCP used in this study) is very plausible. For this reason, the main results



of this study will focus on the RCP8.5/SSP5 combination. However, some analysis will also be performed considering all the SSPs in order to show the differences between the socioeconomic projections (see Section 3.2.3).

### 3.2.3 Future probability of damaging events

Finally, changes in the probability of damaging events for both regions have been assessed when considering a global warming of 1.5, 2 and 3 °C. In this case, only three damage percentiles (50th, 60th, 70th) were selected for defining a "large damaging event". Figure 9 shows the change in this probability for Catalonia, for all the three percentiles of damage and taking into account the mean precipitation recorded in 24 h and the population considering the SSP5 socioeconomic scenario. As can be observed, this probability increases with respect to the reference period (1976-2005) for all the models and warming periods. The increase is higher when greater warming is considered in most of the cases (the increase is usually larger with 3 °C than with 1.5 and 2 °C), which emphasizes the importance of limiting global warming as much as possible. Furthermore, the increase in probability is greater for higher percentiles of damage.

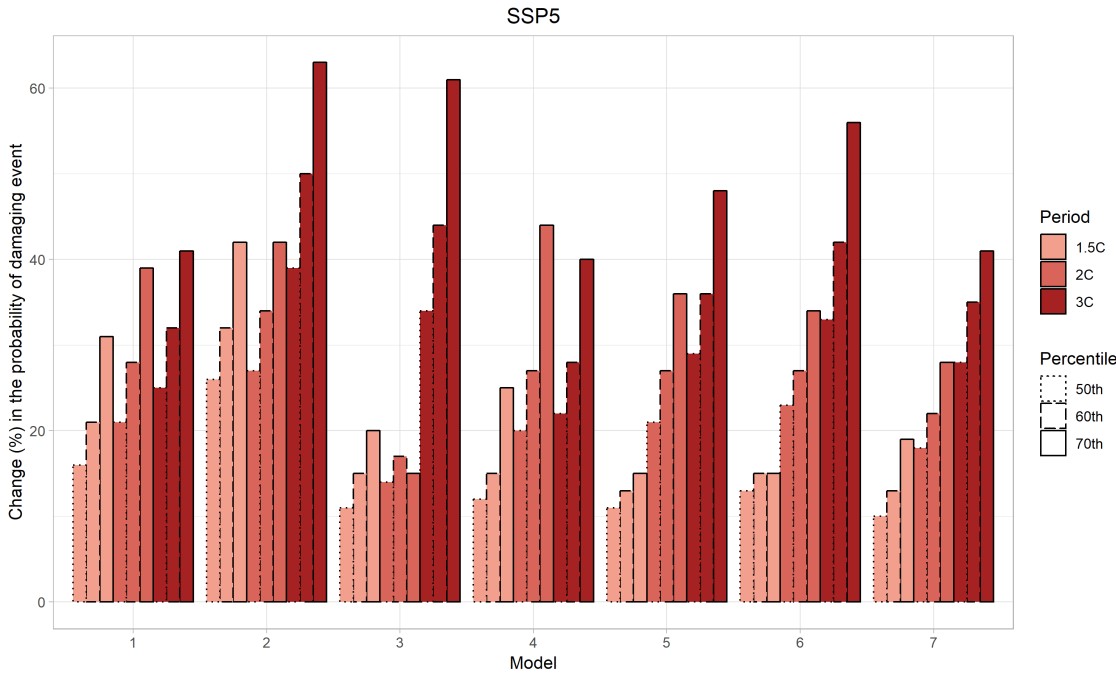

**Figure 9.** Change in the probability of a damaging event in Catalonia with respect to the reference period (1976-2005) for the 50, 60 and 70th damage percentiles and for all models when using the SSP5 socioeconomic scenario. Different hatchings around bars show the different percentiles of damage

The results presented above show the change in probability of damaging events due to both climate change and increasing exposure (i.e. increasing population), both of which have been shown to have a significant relationship with flood damage in Section 3.1.1. In Table 5, the change in the probability of damaging events is shown for the 70th damage percentile, when



keeping population constant at today's levels (left) compared to when using the SSP5 population data (right). The results show that the increase in probability is higher when both climate and population change are included. When population is considered, all the periods and models clearly show a higher increase in the probability of a damaging event, which is statistically significant for all cases. Nevertheless, when mean precipitation accumulated in 24 h is used as the sole independent variable of the model, 5 not all the models show an increase in the probability of damaging events. These results point out the importance of including variables that represent change in both the climate and socioeconomic conditions.

Finally, in order to see the differences between the socioeconomic scenarios (SSPs), the probability of exceeding the 70th percentile of damage in Catalonia for all the periods, models and SSPs is plotted in Figure 10. Results show an increase in the probability of a damaging event in almost all cases, pointing out that even with lower exposure than in the SSP5 scenario, 10 increases in flood damage are evident. The only exception is for Model 3 in combination with the 2 °C warming period and the SSP3 scenario; this SSP shows a marked reduction in population (see Figure 8). The largest increases in probability are found under SSP5, since this has the largest increase in population.

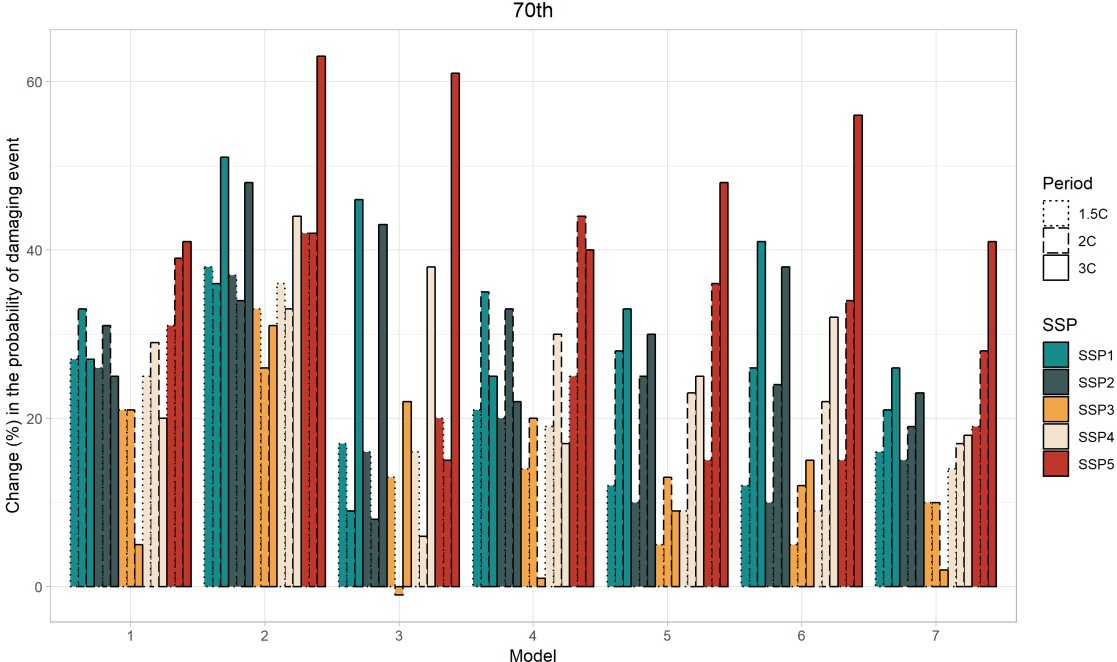

**Figure 10.** Change with respect to the reference period (1976-2005) of exceeding the 70th percentile of damage in Catalonia for all the models, warming periods and socioeconomic scenarios (SSP). Different hatchings around bars show different warming levels

The same analysis has been carried out for the Valencian Community, based on the relationships found for the present climate for this region (see Section 3.1.2). Figure 11 shows the change in the probability of a damaging event (defined by 15 the 3 percentiles of damage) based on the mean precipitation recorded in 24 h and the population (SSP5 scenario). As can be observed, all models and periods show an increase in this probability with respect to the reference period. Therefore, an





increase in the probability of a damaging event occurrence is likely to happen with global warming and with the expected growth of the population, becoming higher when greater warming is considered. Actually, with a global temperature of 3 °C above preindustrial conditions, the change in probability is considerably higher than the other two warming periods (1.5 °C and 2 °C) in most cases, becoming twice as high in some of them. Furthermore, this probability becomes larger for higher

5   percentiles of damages.

The change in the probability of a damaging event when considering the 70th percentile of damage is shown in Table 6. When population is considered constant at today's levels (left), the range of values goes from -4 to 26 %, but only the positive and highest ones are statistically significant. However, when the change in population according to SSP5 is also taken into account (right), all values become positive, greater and most of them statistically significant, highlighting the importance of

10   considering the population while analysing flood damages.

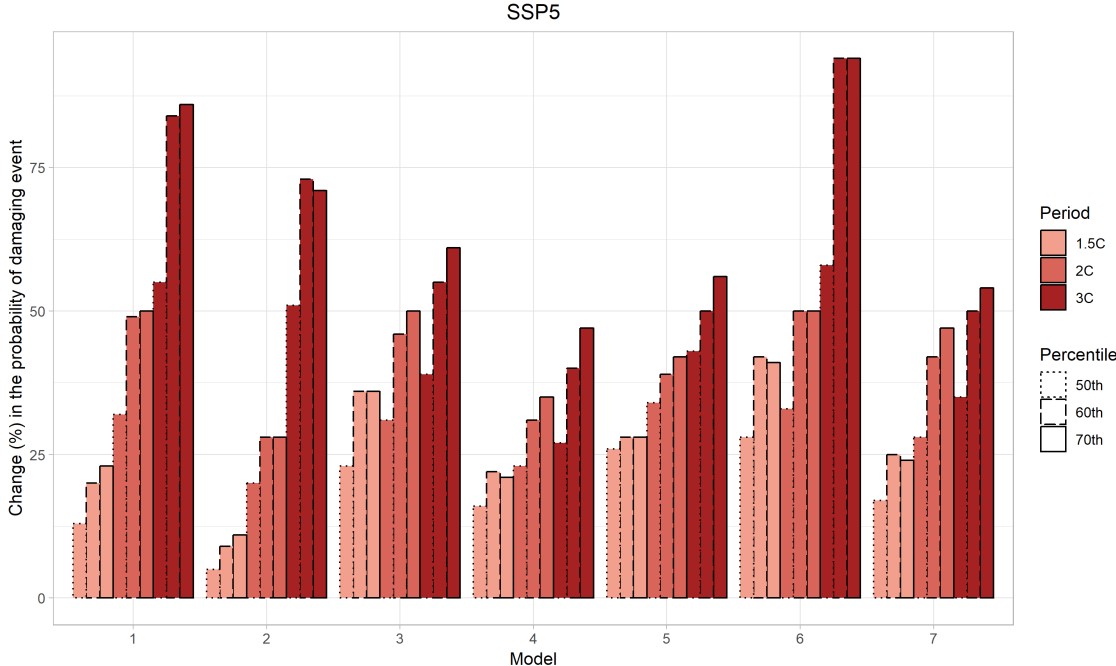

**Figure 11.** Change in the probability of a damaging event in the Valencian Community with respect to the reference period (1976-2005) for the 50, 60 and 70th damage percentiles and for all models when using the SSP5 socioeconomic scenario. Different hatchings around bars show the different percentiles of damage


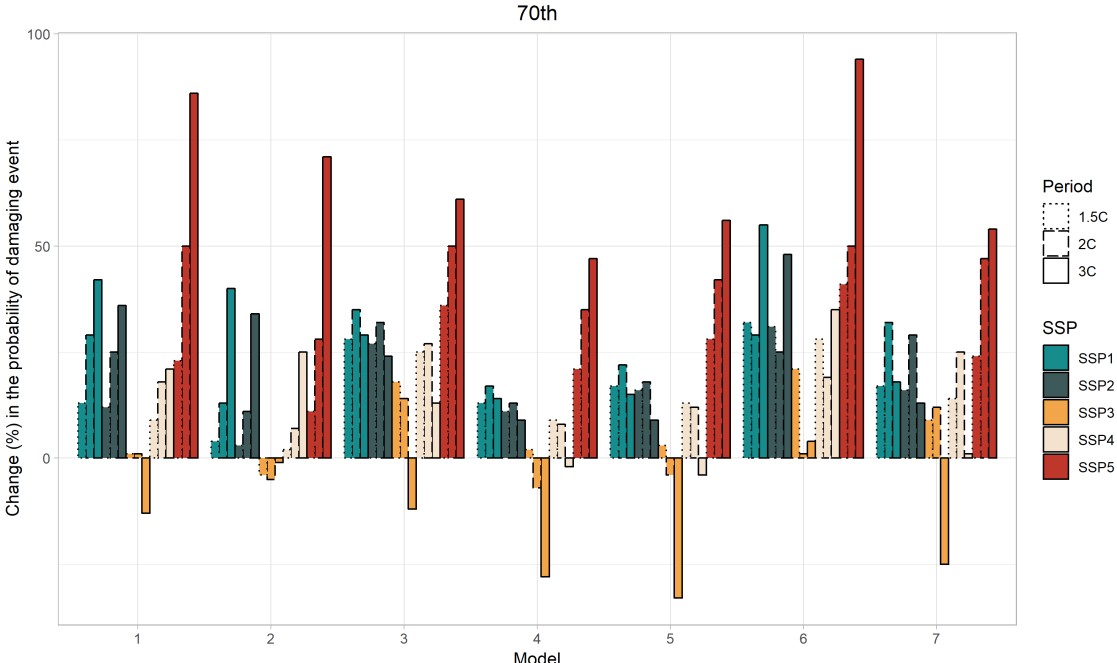

**Figure 12.** Change with respect to the reference period (1976-2005) of exceeding the 70th percentile of damage in the Valencian Community for all the models, warming periods and socioeconomic scenarios (SSP). Different hatchings around bars show different warming levels

The probability of exceeding the 70th percentile of damage for the Valencian Community for all the periods, models, and SSPs is shown in Figure 12. Although most of the cases show an increase in the probability of a damaging event, this increase is not as clear as that of the Catalonia region. When the SSP3 socioeconomic scenario is taken into account, substantial decreases can be found in most models. As in the case of Catalonia, the largest increases are found under the SSP5 scenario, when

higher population growth rates are expected (see Figure 8). These results once again underline the importance of considering socioeconomic changes in flood risk analysis.

Overall, results show a general and marked increase in the probability of a damaging event with global warming with respect to the reference period when considering the RCP 8.5 climate change scenario and the SSP5 socioeconomic scenario. Other

studies, such as Wobus et al. (2014), also show an increase in monetary damages from flooding in nearly all regions of the United States and a total increase in damages by the end of the 21st century of approximately 30 %, although without taking into account other factors involved such as changes in demographics or infrastructure. In the Mediterranean area, other authors found an increase in flood risk associated with extreme precipitation events due to climate change (Cramer et al., 2018; Alfieri et al., 2015a), however the projections in flood hazards in Europe vary a lot regionally (Kundzewicz et al., 2017). Most of the

studies of future flood projections refer to river floods (Alfieri et al., 2015a; Roudier et al., 2016; Rojas et al., 2012), without taking into account other types of floods. Therefore, this study ends with a substantial contribution towards assessing the future





flood damage caused by other types of flood events, such as those caused by heavy precipitation episodes (surface water floods), and also taking into account the changes in the population in the analysis.

## 4 Limitations and future research

The main contribution of this study is the assessment of the probability of future damaging events in two Western Mediterranean
regions, considering both climate change and changes in exposure according to different socioeconomic scenarios. Although the results show a promising methodology for predicting flood damage, several limitations should be taken into account.

Firstly, it should be noted that flood risk analysis is very complex and needs to be treated from a holistic perspective using techniques that take into account all the factors involved, both those related to the hazard of the phenomenon and those related to exposure of the territory. In this study only precipitation and population were taken into account, and although the results
show a good performance of the model for simulating the probability of a damaging event, other variables such as soil physical conditions or preventive measures should be considered. For instance, in Garrote et al. (2016) different simulation scenarios were defined considering the modifications to the terrain due to the construction of fluvial defence structures in the area. Future research should focus on incorporating further variables into the model to reproduce the complexity of flood risk. Furthermore, the present study only focuses on daily precipitation data, while several studies point out the possible better relationship found
between sub-daily data and insurance data in the case of surface water floods (Spekkers et al., 2013; Torgersen et al., 2015; Cortès et al., 2018). Nevertheless, the analysis of sub-daily extremes would require convection-permitting regional climate models (Tramblay and Somot, 2018), and studies such as Beranová et al. (2018) have shown that convection is not capture realistically. Therefore, projected changes of sub-daily precipitation extremes in climate change scenarios based on RCMs not resolving convection need to be interpreted with caution (Beranová et al., 2018). Finally, an effort has to be made to
incorporate indirect and non-tangible damages in flood damage analysis, as these are crucial for evaluating the full impacts caused by natural hazards (Petrucci and Llasat, 2013).

Despite these limitations, this work has provided a first assessment of the link between precipitation and flood damage in a Mediterranean region, also considering the exposure of the territory with the incorporation of the population in the model and the impacts of climate change projections.

## 25 5 Conclusions

The NW Mediterranean region experiences heavy precipitation every year and flash floods that occasionally produce catastrophic damages (Llasat et al., 2013). Both of the western Mediterranean regions covered in this study (Catalonia and the Valencian Community) are prone to these events, most of them caused by local heavy precipitation events (Llasat et al., 2016).

In this study, the relationship between heavy precipitation and flood damage for both regions has been analysed. Generalized
Linear Mixed Models have been used in order to estimate the probability of damaging events taking into account both hazard and exposure. The results show that the probability of a damaging event increases with precipitation and population of the





basin. For both regions of study the values under the ROC curve (RA) indicate that our model has a good performance, with values close to 1 in all cases. These results improve those obtained by using a simple logistic model with precipitation recorded in 24 h as the only explanatory variable (see Cortès et al., 2018), pointing out the need of incorporating variables such as population that give information about the exposure of the territory for explaining flood damage.

Using the models developed in this study, we also assessed the future probability of damaging flood events under scenarios of climate and population change. We show that this probability increases with respect to the reference period (1976-2005) for most models and warming periods in both regions when considering the RCP8.5/SSP5 combination, being higher in the case of Catalonia. Furthermore, a remarkable result found in both areas is that this change is usually larger when greater warming is considered. This points out the importance of limiting the global warming to the lowest possible levels with mitigation

actions. A reduction in the probability of damaging events is only found when considering SSP3 (and in some cases SSP4 for the Valencian Community), since this scenario shows a marked decrease in the population by the end of 21st century in both areas. Furthermore, in both regions the results show that the increase in probability is higher when both climate and population change are included, highlighting the importance of considering variables that take into account the exposure of the territory in flood damage analysis.

To summarise, we have estimated the changes in the probability that a flood event causing large damage will occur, based on the changes in precipitation expected with global warming, and also considering the expected changes in demographics. The general increase found in the probability of damaging event should be taken into account in flood management strategies. Therefore, our results suggest the urgency to develop and apply adaptation and mitigation strategies, also considering that many areas in Spain already suffer from problems related to climate change (Quiroga et al., 2011; Turco and Llasat, 2011).





# Appendix A: Methods

## A1    Future probability of flood damage

### A1.1    Population projections

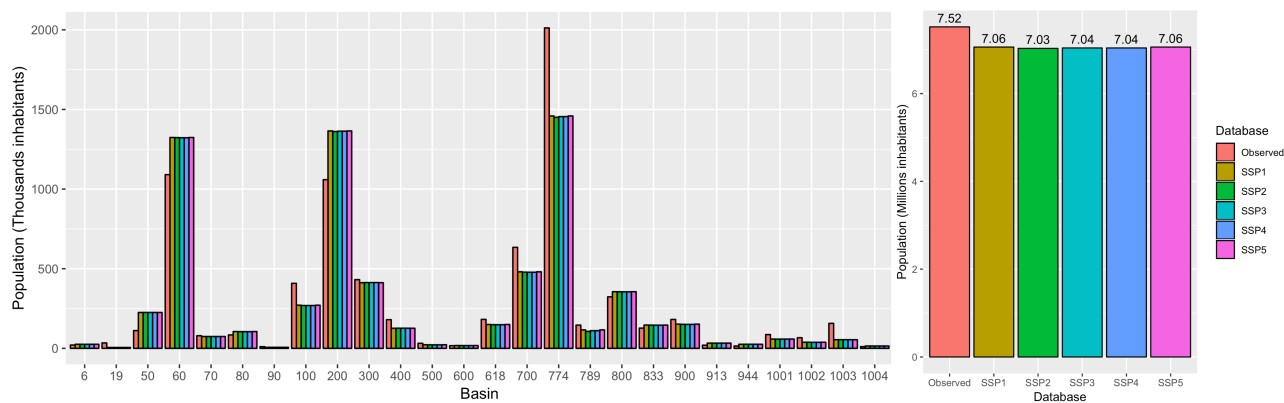

**Figure A1.** Population (year 2010) for each basin of Catalonia (left) and for the whole region (right) for the different databases. The numbers on the x-axis indicate the code of each basin (see Table A1)



| Code | Name of the basin |
|------|-------------------|
| 6 | El Daró |
| 19 | Riera de Riudecanyes |
| 50 | La Tordera |
| 60 | El Besòs |
| 70 | El Foix |
| 80 | El Francolí |
| 90 | Eth Garona |
| 100 | El Ter |
| 200 | El Llobregat |
| 300 | El Segre |
| 400 | L'Ebre |
| 500 | Rieres Costa Brava Nord |
| 600 | Rieres Costa Brava Centre |
| 618 | Rieres Costa Brava Sud |
| 700 | Rieres del Maresme |
| 774 | Torrents de l'Àrea Metropolitana de Barcelona |
| 789 | Rieres litorals Llobregat |
| 800 | Rieres del Garraf |
| 833 | Rieres Tarragona Nord |
| 900 | Rieres Tarragona Sud |
| 913 | Rieres Meridionals de Tarragona |
| 944 | Rieres litorals Ebre Nord |
| 1001 | El Tec; Rieres litorals Muga; La Muga |
| 1002 | Rieres litorals Fluvià; El Fluvià |
| 1003 | El Gaià; Rieres Tarragona Centre |
| 1004 | La Sénia; Rieres litorals Ebre Sud |

**Table A1.** Corresponding names of the codes of the basins of Catalonia region





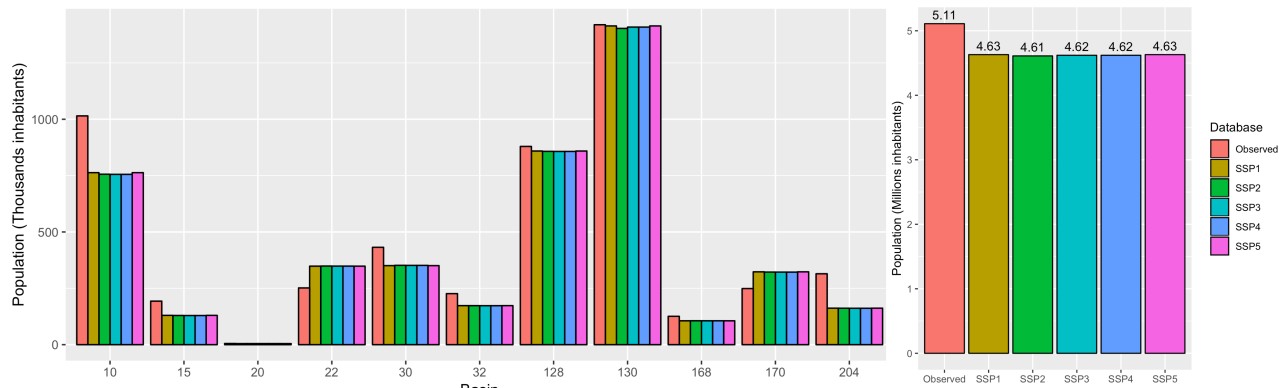

**Figure A2.** Population (year 2010) for each basin of the Valencian Community (left) and for the whole region (right) for the different databases. The numbers on the x-axis indicate the code of each basin (see Table A2)

| Code | Name of the basin |
|------|-------------------|
| 10   | Vinalop-Alacant |
| 15   | Marina Baixa |
| 20   | Ebre |
| 22   | Marina Alta |
| 30   | Segura |
| 32   | Serpis |
| 128  | Jucar |
| 168  | Palancia-Los Valles |
| 170  | Mijares-Plana de Castelló |
| 204  | Cenia-Maestrazgo |

**Table A2.** Corresponding names of the codes of the basins of the Valencian Community region





# Appendix B: Results

## B1 Present climate

### B1.1 Catalonia

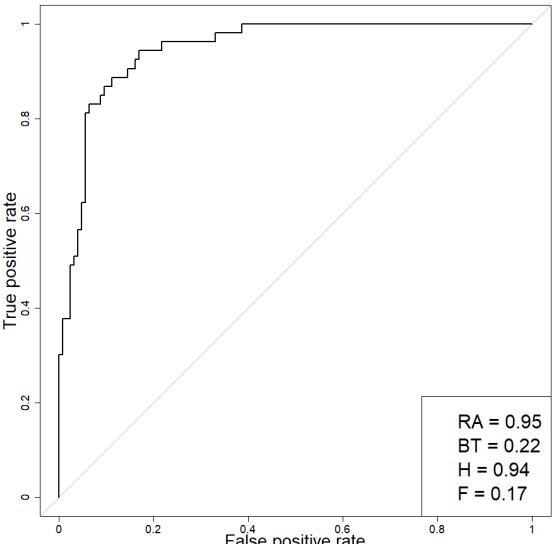

**Figure B1.** Relative operating characteristic (ROC) diagram for above 70th percentile of damage predictions using Eq. 4 for Catalonia ($P_0$=40 mm/24 h). Each value of the ROC curve indicates a set of probability forecasts by stepping a decision threshold with 1 % probability through the modelling results. The numbers inside the plot are the ROC area (RA) and the best threshold (BT), here defined as the threshold that maximises the difference between the hit rate (H) and the false alarm rate (F)





## B1.2 Valencian Community

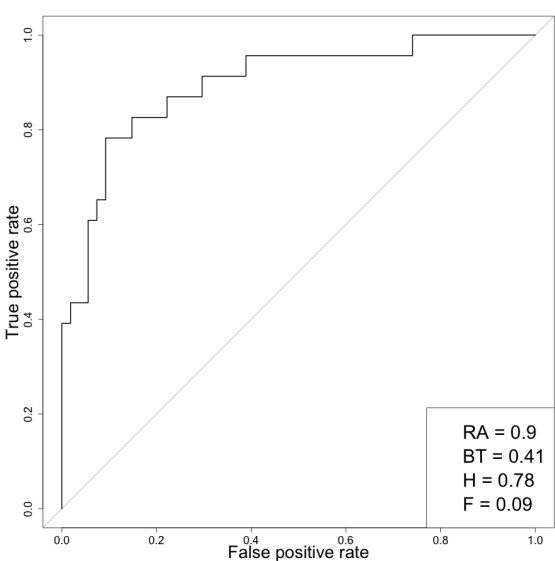

**Figure B2.** Relative operating characteristic (ROC) diagram for above 70th percentile of damage predictions using Eq. 5 for the Valencian Community ($P_0$=40 mm/24 h). Each value of the ROC curve indicates a set of probability forecasts by stepping a decision threshold with 1 % probability through the modelling results. The numbers inside the plot are the ROC area (RA) and the best threshold (BT), here defined as the threshold that maximises the difference between the hit rate (H) and the false alarm rate (F)

*Competing interests.* The authors declare that they have no conflict of interest

5 *Acknowledgements.* This work has been supported by the Spanish project M-COSTADAPT (CTM2017-83655-C2-R) of the FEDER/Ministry of Science, Innovation and Universities - AEI, for the project EFA210/16/ PIRAGUA, INTERREG POCTEFA 2014-2020 and for the Water Research Institute (IdRA) of the University of Barcelona. It was conducted under the framework of the HyMeX Programme (HYdrological cycle in the Mediterranean EXperiment) and the Panta Rhei WG Changes in Flood Risk. P.J.Ward received funding from the Netherlands Or-



ganisation for Scientific Research (NOW) in the form of VIDI grant 016.161.324. M.Turco has received funding from the European Union's Horizon 2020 research and innovation programme under the Marie Skłodowska-Curie grant agreement No. 740073 (CLIM4CROP project).





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



| Model | Institute | GCM | RCM | 1.5 °C | 2 °C | 3 °C |
|---|---|---|---|---|---|---|
| 1 | KNMI | EC-EARTH | RACMO22E | 2017–2046 | 2032–2061 | 2055–2084 |
| 2 | SMHI | HadGEM2-ES | RCA4 | 2011–2040 | 2023–2052 | 2041–2070 |
| 3 | SMHI | EC-EARTH | RCA4 | 2014–2043 | 2028–2057 | 2053–2082 |
| 4 | MPI-CSC | MPI-ESM-LR | REMO2009 | 2017–2046 | 2031–2060 | 2054–2083 |
| 5 | CLMcom | MPI-ESM-LR | CCLM4-8-17 | 2017–2046 | 2031–2060 | 2054–2083 |
| 6 | SMHI | MPI-ESM-LR | RCA4 | 2017–2046 | 2031–2060 | 2054–2083 |
| 7 | CLMcom | EC-EARTH | CCLM4-8-17 | 2014–2043 | 2028–2057 | 2053–2082 |

**Table 1.** EURO-CORDEX climate models used, their characteristics and the periods of each GCM for the three warming levels considered in this study





|  | Observed | |
|  | Yes | No |
| --- | --- | --- |
| **Forecast** Yes | a | b |
| **Forecast** No | c | d |

**Table 2.** Contingency table to support Eq.2 and Eq.3




| Percentile | $\beta_0$ | $\beta_1$ | $\beta_2$ | RA |
|---|---|---|---|---|
| 10 | -14.74 | 3.05 | 0.94 | 0.98 |
| 20 | -12.98 | 1.60 * | 0.72 | 0.94 |
| 30 | -14.73 | 2.16 | 0.64 | 0.95 |
| 40 | -24.87 | 3.12 | 1.11 | 0.97 |
| 50 | -18.46 | 2.17 | 0.81 | 0.96 |
| 60 | -17.74 | 2.10 | 0.69 | 0.95 |
| 70 | -20.92 | 2.98 | 0.57 | 0.95 |
| 80 | -27.22 | 4.19 | 0.54 | 0.96 |
| 90 | -21.46 | 3.28 | 0.31 * | 0.95 |

**Table 3.** Parameters of the Generalized Linear Mixed Model and the area under the ROC curve (RA) values for all the percentiles of damage for Catalonia. $\beta_0$, $\beta_1$ and $\beta_2$ represent the regression coefficients of the model (intercept, precipitation, and population respectively). * shows the non-significant regression coefficients (p value > 0.05)



| Percentile | $\beta_0$ | $\beta_1$ | $\beta_2$ | RA |
|---|---|---|---|---|
| 10 | -183.57 | 34.47 | 4.95 | 1.00 |
| 20 | -188.88 | 32.58 | 5.53 | 1.00 |
| 30 | -43.83 | 4.59 | 1.99 | 1.00 |
| 40 | -42.40 | 3.14 | 2.31 | 0.96 |
| 50 | -42.85 | 3.06 | 2.32 | 0.90 |
| 60 | -60.63 | 4.46 | 3.16 | 0.95 |
| 70 | -40.96 | 2.33 | 2.32 | 0.90 |
| 80 | -49.60 | 3.80 | 2.41 | 0.89 |
| 90 | -50.97 | 4.05 | 2.34 | 0.94 |

**Table 4.** Parameters of the Generalized Linear Mixed Model and the area under the ROC curve (RA) values for all the percentiles of damage for the Valencian Community. $\beta_0$, $\beta_1$ and $\beta_2$ represent the regression coefficients of the model (intercept, precipitation, and population respectively). In this case, all the regression coefficients are statistically significant (p value < 0.05)





| Model | Precipitation | | | Precipitation + population | | |
|---|---|---|---|---|---|---|
| | 1.5 °C | 2 °C | 3 °C | 1.5 °C | 2 °C | 3 °C |
| 1 | 13 % | 15 % | 12 % | 31 % | 39 % | 41 % |
| 2 | 11 % | 10 % | 12 % | 42 % | 42 % | 63 % |
| 3 | 6 % | -2 % | 17 % | 20 % | 15 % | 61 % |
| 4 | 11 % | 16 % | 9 % | 25 % | 44 % | 40 % |
| 5 | 3 % | 10 % | 10 % | 15 % | 36 % | 48 % |
| 6 | 1 % | 8 % | 14 % | 15 % | 34 % | 56 % |
| 7 | 6 % | 5 % | 4 % | 19 % | 28 % | 41 % |

**Table 5.** Change in the probability of a damaging event in Catalonia for the 70th damage percentile and the three warming periods with the mean 24 h precipitation as explanatory variable (left) and mean 24 h precipitation and population (SSP5) as explanatory variables (right). The gray cells show statically significant (p value < 0.05) increases (light) and decreases (dark) in relation to the reference period (1976-2005)





| Model | Precipitation | | | Precipitation + population | | |
|---|---|---|---|---|---|---|
| | 1.5 °C | 2 °C | 3 °C | 1.5 °C | 2 °C | 3 °C |
| 1 | 12 % | 12 % | 16 % | 23 % | 50 % | 86 % |
| 2 | 4 % | 2 % | 8 % | 11 % | 28 % | 71 % |
| 3 | 13 % | 19 % | 14 % | 36 % | 50 % | 61 % |
| 4 | 2 % | 1 % | 3 % | 21 % | 35 % | 47 % |
| 5 | -4 % | -2 % | -4 % | 28 % | 42 % | 56 % |
| 6 | 16 % | 7 % | 26 % | 41 % | 50 % | 94 % |
| 7 | 10 % | 20 % | 9 % | 24 % | 47 % | 54 % |

**Table 6.** Change in the probability of a damaging event in the Valencian Community for the 70th damage percentile and the three warming periods with the mean 24 h precipitation as explanatory variable (left) and mean 24 h precipitation and population (SSP5) as explanatory variables (right). The gray cells show statically significant (p value < 0.05) increases (light) and decreases (dark) in relation to the reference period (1976-2005)