# Peer review of "Changes in flood damage with global warming in the east coast of Spain"

_Natural Hazards and Earth System Sciences, 2019_

## Referee Comment (RC1) · Alfredo Perez (Referee) · 30 Aug 2019

GENERAL COMMENTS:

In this article the authors carry out an assessment of the probability of future damaging events in two Western Mediterranean regions in Spain, considering both climate change and changes in exposure according to different socioeconomic scenarios. Although the results show a promising methodology for predicting flood damage, several suggestions should take into account before the work will be publish.

A)

1. Introduction.

[Figure]

"In Spain, the findings of Barredo et al. (2012) align with these results; they find no significant trend in adjusted insured flood losses between 1971 and 2008. These studies show the need to include exposure and vulnerability changes in future risk projections, which clearly contribute substantially to changing risks."

I think that relevance given to social variables (exposure and vulnerability) in this work is very correct. In recent years, they have acquired as much or more importance than the physical ones within the risk formula. However, from my point of view, according to the results and the significance of these social variables in the models obtained, these social variables should be better explained. Therefore, I would like to see a paragraph in the introduction of the new version of the work that deepens in this regard. Some bibliographical references such as the following could be useful:

López-Martínez, F.; Gil-Guirado, S. y Pérez-Morales, A. (2017). ÂńWho can you trust? Implications of institutional vulnerability in flood exposure along the Spanish Mediterranean coastÂż. Environmental Science & Policy, 76, 29-39. 10.1016/j.envsci.2017.06.004

Pérez-Morales, A., Gil-Guirado, S., and Olcina-Cantos, J., 2018. Housing bubbles and the increase of flood exposure. Failures in flood risk management on the Spanish south-eastern coast (1975–2013). Journal of Flood Risk Management, 11 (S1), S302–S313. 10.1111/jfr3.2018.11.issue-S1

Raschky, P.A., 2008. Institutions and the losses from natural disasters. Nat. Hazards Earth Syst. Sci. 8 (4), 627–634. http://dx.doi.org/10.5194/nhess-8-627-2008.

Fekete, A. Int J Disaster Risk Sci (2019) 10: 220. https://doi.org/10.1007/s13753-019-0213-1

B)

4. Limitations and future research.

"Future research should focus on incorporating further variables into the model to reproduce the complexity of flood risk. flood damage caused by other types of flood events, such as those caused by heavy precipitation episodes (surface water floods), and also taking into account the changes in the population in the analysis"

On the other hand, I suggest to the authors that, for the improvement of the models in future works, take into account those variables more related to building. Cadastre offers high-resolution temporal space data very useful in this regard. Although variables such as the number of inhabitants, population density per square kilometer, etc. can be significant, they are frankly improvable due to their level of temporal space aggregation. This implies an important generalization that does not correctly represent the exposure, much less vulnerability. In the case of flooding hazard, cadastre and the variable "number of buildings per census tract" of the INE are much more precise. In fact, Cadastre is much more related to the database of damages registered by the insurance consortium, since these indemnities are associated with insurance policies connected to buildings or homes with cadastral references.

C)

2.2 Data. Line 21. "The population data corresponds to the year when the flood event took place".

According to this, the availability and source consulted year by year must be indicated. Municipal Register of inhabitants or Population Census.

D)

2.3.1 Generalized Linear Mixed Model.

As far as my knowledge on the subject goes, it would be convenient to carry out a spatial autocorrelation test (Moran's I) to rule out the assumptions discussed in the text of the paper. Likewise, it would be convenient to compare the accuracy of the results with other models such as Geographically Weighted Logistic Regression (LGWR) since, as indicated in the work, there is very strong spatial autocorrelation that could reduce the

accuracy of the results.

The LGWR models are effective to solve spatial autocorrelation and non-stationarity, or regional variation, of some variables. Thus, the results of the GLMM models could be improved since it is possible to differentiate the local spatial variations of the parameters estimated by means of the implementation of a kernel function, that allows to make estimations adjusted to each observation giving greater influence on the closer observations.

E)

3.2.3 Future probability of damaging events. Regional rivalry (SSP3)

Population scenarios and, specifically SSP3, which represents a decrease in damage should be explained in greater detail by relating the comment within the context of the study area. In fact, this aspect is crucial, since it is a good find from which to establish these adaptation measures or strategies to climate change impacts.

SPECIFIC ASPECTS:

A) Figure 2. Map of both regions of study (put white lines of the provinces)

B) "The solid line indicates the best estimate while the shaded blue bands indicate the 95 % confidence interval". This phrase is repeated both in the body text and in the figures caption. Consider removing from the body text.

---

## Referee Comment (RC2) · Anonymous Referee #2 · 10 Oct 2019

The paper presents an analysis of the expected change of flood damages under future climate scenarios for two regions on the North-East of the Iberian Peninsula. The authors present a model to estimate the probability of occurrence of damaging events based on daily precipitation and population. The model is calibrated for observed events in the two regions and then it is used to estimate changes in the probability of occurrence of damaging events with a global warming of 1.5, 2 and 3 °C and population estimates consistent with SSPs. The topic is relevant because the scientific community is currently addressing flood risk changes under climate change incorporating exposure. In my opinion, the main contributions of the paper are the application of the Generalized Linear Model to obtain the probability of a damaging event and the set of results obtained for the two regions under different scenarios. I also found

interesting the discussion on the role of explanatory variables (precipitation and population) on determining the flood risk. These results are relevant for scientists working on climate risks and the methodology can be extended to other domains.

Overall, the manuscript is correctly organized and well written, adequately illustrated with figures and tables. The topic fits well within the scope of Natural Hazards and Earth System Sciences, the objectives are clear and well identified and the conclusions are adequately supported by the results and discussion. Therefore, I think the paper deserves publication in NHESS. I am just offering a few suggestions for minor revisions that could improve the manuscript with a little further work.

In my opinion the presentation of the formulation and validation of the GLM model could be improved with some additional explanations. I found the problem formulation a bit confusing. The authors mention "the probability of large damaging events occurring given a certain precipitation amount". The "large damaging event" is related to a certain quantile of the sample of insurance compensations. I understood that the "certain precipitation amount" is 40 mm in 24 h. Therefore, the model estimates the probability of having damages exceeding the threshold when mean precipitation exceeds 40 mm in 24 h. However, as shown in the appendix, basin populations are very different and damages should be evaluated according to the population. How do you account for the fact that basins are heterogeneous in size and have different population densities? Do you assume that the entire basin population is affected by the flood? The second question is related to model validation through the ROC diagram. I got the impression that the same sample was used for model fitting and for model validation. Could the authors please clarify this point? The third question is related to model application in climate scenarios. The results shown correspond to changes in the probability of damaging events. How are those changes computed? The GLM produces the probability of having a "damaging" event, given a precipitation amount P and a basin population R. Therefore, it produces one probability per event. Since the number and nature of events are different in the control and in the future periods, how are the changes in

probability computed using the GLM? I would appreciate if the authors could elaborate on this, since the Methodology section ends abruptly after the presentation of precipitation and population projections.

Regarding the climate projections, the authors mention that they selected 30-year periods of EURO-CORDEX simulations starting from the year when the 20-year running mean exceeds the temperature thresholds. These periods are shown on Table 1. However, the SSP population projections are time dependent, but not temperature dependent. Where the analyses made with a different population for each model? Is this a methodological inconsistency? Could the authors provide a brief discussion on this?

Regarding data, the authors mention several data sources to identify flood events in the two regions (INUNGAMA, PRESSGAMA and FLOODHYMEX). They seem to use the events identified in these datasets to obtain the damage data provided by the Insurance Compensation Consortium, with a continuous record 1996-2015. Did you check if there are events with relevant damage data in the ICC dataset not included in the other data sources?

Apart from the above points, there are a few practical details that could improve the paper:

Pag 3, line 14, "summarises". . should be "summarise"? Pag 13, lines 2-3, "which was affected by 69 flood events between 1996 and 2015, resulting in 171 flood cases". .Which is the difference between a "flood event" and a "flood case"? Pag 16, line 15, "showed". . . should be "shown"? Pag 22, line 17, "capture". . . should be "captured"? Pag 26, Basin 130 is missing from the list in Table A2.

---

## Author Comment (AC1) · 20 Nov 2019

Reviewer #1.  GENERAL COMMENTS: In this article the authors carry out an assessment of the probability of future damaging events in two Western Mediterranean regions in Spain, considering both climate change and changes in exposure according to different socioeconomic scenarios. Although the results show a promising methodology for predicting flood damage, several suggestions should take into account before the work will be publish.

Response: We would like to thank the reviewer for his very constructive comments.

Reviewer #1. A) Introduction: "In Spain, the findings of Barredo et al. (2012) align with these results; they find no significant trend in adjusted insured flood losses between

1971 and 2008. These studies show the need to include exposure and vulnerability changes in future risk projections, which clearly contribute substantially to changing risks." I think that relevance given to social variables (exposure and vulnerability) in this work is very correct. In recent years, they have acquired as much or more importance than the physical ones within the risk formula. However, from my point of view, according to the results and the significance of these social variables in the models obtained, these social variables should be better explained. Therefore, I would like to see a paragraph in the introduction of the new version of the work that deepens in this regard.

Response: We found the bibliography provided by the reviewer very interesting, and we agree on including a paragraph in the introduction section that goes deeper in the role of exposure and vulnerability factors in flood risk. We have added this paragraph (line 26, page 2): "Other studies such as López-Martínez et al. (2017) and Pérez-Morales et al. (2018) also mention the vulnerability and exposure components as possible drivers responsible for the increasing flood risk in Mediterranean regions of Spain. Specifically, these studies mention the institutional vulnerability (Raschky et al. 2008), which represents the sensitivity of public administrations to deal with hazards, as one of the main causes. Also, Pérez-Morales et al. (2018) demonstrated that the exposure in flood-prone areas in the south-east of Spain (part of the region of study) has increased in the last decades, due to a poor management of these areas by government institutions and the regulation adopted by them."

Reviewer #1. B) Limitations and future research: "Future research should focus on incorporating further variables into the model to reproduce the complexity of flood risk. flood damage caused by other types of flood events, such as those caused by heavy precipitation episodes (surface water floods), and also taking into account the changes in the population in the analysis". On the other hand, I suggest to the authors that, for the improvement of the models in future works, take into account those variables more related to building. Cadastre offers high-resolution temporal space data very

useful in this regard. Although variables such as the number of inhabitants, population density per square kilometre, etc. can be significant, they are frankly improvable due to their level of temporal space aggregation. This implies an important generalization that does not correctly represent the exposure, much less vulnerability. In the case of flooding hazard, cadastre and the variable "number of buildings per census tract" of the INE are much more precise. In fact, Cadastre is much more related to the database of damages registered by the insurance consortium, since these indemnities are associated with insurance policies connected to buildings or homes with cadastral references.

Response: We wish to thank the reviewer for this constructive comment and for the information provided. We agree on the improvement of the future works by considering more variables related to the exposure and vulnerability when analysing flood risk. In this work, we have chosen representative exposure variable to obtain in future projections in order to use the same model developed in the present climate for predicting future flood damage. In order to emphasise this, we have changed the following sentence in the manuscript (line 13, page 23): "Future research should focus on incorporating further variables into the model to reproduce the complexity of flood risk, not only regarding hazard drivers but also considering more variables related to the exposure and vulnerability, as for example those related to the buildings present in the study region."

Reviewer #1. C) Data. Line 21: "The population data corresponds to the year when the flood event took place". According to this, the availability and source consulted year by year must be indicated. Municipal Register of inhabitants or Population Census.

Response: We agree with the reviewer and we have added this information in the manuscript. In addition, the reviewer's comment has helped us to realise that there was an error in the population data source of Catalonia. Population data of both regions (Catalonia and the Valencian Community) have been obtained from the same source (Spanish National Statistics Institute). We have changed the sentence as following

(line 19, page 6): "Population data for both regions were obtained from the municipal register of inhabitants provided by the Spanish National Statistics Institute..."

Reviewer #1. D) Generalized Linear Mixed Model: As far as my knowledge on the subject goes, it would be convenient to carry out a spatial autocorrelation test (Moran's I) to rule out the assumptions discussed in the text of the paper.

Response: The spatial autocorrelation is almost negligible in this case. The model includes a random effect associated with the geographical basin to reflect the auto-correlation for observations coming from the same basin, and the estimates for this component have a standard deviation that can be considered zero. This means that when considering the observations from the same basin, these observations can be considered independent. For analysing the case of spatial correlation as a function of the distance between basins, we have applied the suggested test (Moran's I) for three variables: Number of events, sum of damages and mean damage. The p-values for the test in each variable confirm that there is no need to consider spatial correlation in this case. In order to clarify this point and to not create misunderstanding, we have changed the following sentence in the manuscript (page 8, line 16): "This implies high correlations within the observations, not guaranteeing the independence requirement of the Generalized Linear Models (GLM)."

For: "This implies that the observations are grouped by random factors, not guarantee-ing the independence requirement of the Generalized Linear Models (GLM)."

Reviewer #1. D) Generalized Linear Mixed Model: Likewise, it would be convenient to compare the accuracy of the results with other models such as Geographically Weighted Logistic Regression (LGWR) since, as indicated in the work, there is very strong spatial autocorrelation that could reduce the accuracy of the results. The LGWR models are effective to solve spatial autocorrelation and non-stationarity, or regional variation, of some variables. Thus, the results of the GLMM models could be im-proved since it is possible to differentiate the local spatial variations of the parameters
estimated by means of the implementation of a kernel function, that allows to make estimations adjusted to each observation giving greater influence on the closer observations.

Response: We agree with the reviewer that if there would have been significant the spatial autocorrelation, these kind of models could have been useful. But in this case, there are two random factors that are crossed: basin (spatial effect) and flood events (time effect). The model estimation indicates that the first is not significant but the second is. We have fitted the proposed models (LGWR) to evaluate the impact of spatial autocorrelation in the parameter estimation, but, to our knowledge, the implementation (https://github.com/pysal/mgwr) does not include the existence of random effects, so the time effect is not included in this model, which is clearly significant in our case. The results of the model obtained with LGWR are the same as the GLMM when the flood event effect is not considered.

Reviewer #1. E) Future probability of damaging events. Regional rivalry (SSP3): Population scenarios and, specifically SSP3, which represents a decrease in damage should be explained in greater detail by relating the comment within the context of the study area. In fact, this aspect is crucial, since it is a good find from which to establish these adaptation measures or strategies to climate change impacts.

Response: We agree with the reviewer's comment and we have included a paragraph in the results section (line 11, page 19): "This scenario refers to a fragmented world with an emphasis on security at the expense of international development. In rich-OECD countries (defined by OECD membership and the World Bank category of high-income country), to which both regions of our study belong, the SSP3 scenario depicts a low fertility rate, high mortality and low immigration (Samir and Lutz, 2017). Therefore, population is assumed to decrease in these countries (see Figure 8). On the other hand, the largest increases in probability are found under SSP5, since this has the largest increase in population (see Figure 8). As mentioned before, this scenario refers to a world that emphasizes technological progress and where economic growth

is fostered by the rapid development of human capital. In rich-OECD countries a low mortality rate, high immigration and a relatively high fertility rate are expected, due to a high technology and a very high standard of living that allows for easier combination of work and family (Samir and Lutz, 2017)."

Reviewer #1. SPECIFIC ASPECTS: A) Figure 2. Map of both regions of study (put white lines of the provinces)

Response: We have followed the reviewer's suggestion and tried to include provinces lines in the map but for visual criteria we have decided to consider only basin borders.

B) "The solid line indicates the best estimate while the shaded blue bands indicate the 95 % confidence interval". This phrase is repeated both in the body text and in the figures caption. Consider removing from the body text.

Response: We have removed from the body text following the reviewer's suggestion.
* * *

---

## Author Comment (AC2) · 20 Nov 2019

Reviewer #2. The paper presents an analysis of the expected change of flood damages under future climate scenarios for two regions on the North-East of the Iberian Peninsula. The authors present a model to estimate the probability of occurrence of damaging events based on daily precipitation and population. The model is calibrated for observed events in the two regions and then it is used to estimate changes in the probability of occurrence of damaging events with a global warming of 1.5, 2 and 3 C and population estimates consistent with SSPs. The topic is relevant because the scientific community is currently addressing flood risk changes under climate change incorporating exposure. In my opinion, the main contributions of the paper are the application of the Generalized Linear Model to obtain the probability of a damaging

event and the set of results obtained for the two regions under different scenarios. I also found interesting the discussion on the role of explanatory variables (precipitation and population) on determining the flood risk. These results are relevant for scientists working on climate risks and the methodology can be extended to other domains. Overall, the manuscript is correctly organized and well written, adequately illustrated with figures and tables. The topic fits well within the scope of Natural Hazards and Earth System Sciences, the objectives are clear and well identified and the conclusions are adequately supported by the results and discussion. Therefore, I think the paper deserves publication in NHESS. I am just offering a few suggestions for minor revisions that could improve the manuscript with a little further work.

Response: We wish to thank the anonymous referee for his/her useful and constructive comments. Each specific point has been addressed in the manuscript as explained in the following document.

Reviewer #2. In my opinion the presentation of the formulation and validation of the GLM model could be improved with some additional explanations. I found the problem formulation a bit confusing. The authors mention "the probability of large damaging events occurring given a certain precipitation amount". The "large damaging event" is related to a certain quantile of the sample of insurance compensations. I understood that the "certain precipitation amount" is 40 mm in 24 h. Therefore, the model estimates the probability of having damages exceeding the threshold when mean precipitation exceeds 40 mm in 24 h. However, as shown in the appendix, basin populations are very different and damages should be evaluated according to the population. How do you account for the fact that basins are heterogeneous in size and have different population densities? Do you assume that the entire basin population is affected by the flood?

Response: As the reviewer correctly mentioned, the model estimates the probability of a damaging event, defined as the exceedance of different damage thresholds, when mean precipitation exceeds 40 mm in 24 h. In the model, we take into account the

heterogeneities of the basin incorporating the population as explanatory variable. As the reviewer commented, we assume that the entire population of the basin is affected by the flood event, since the smallest unit of study is the basin. We also considered the area of the basin as explanatory variable, but it didn't provide information on the response variable. We have added a new sentence in the following paragraph of the article to clarify this point (line 31-32, page 7): "The minimum geographical unit of the study is the river-basin-scale and we only consider the flood cases that recorded a mean precipitation in the basin higher than 40 mm in 24 h. Therefore, the model estimates the probability of having economic damages exceeding different damage thresholds (percentiles) when mean precipitation in the basin exceeds 40 mm in 24 h."

Reviewer #2. The second question is related to model validation through the ROC diagram. I got the impression that the same sample was used for model fitting and for model validation. Could the authors please clarify this point?

Response: In this case, the out-of-sample validation has not been included in the study, only the in-sample validation. So the reviewer is right: we use the whole data to estimate the model and validate the predictions. When dealing with mixed models, the cross-validation can be a bit more complicated, because of the grouped structure of the data. In this study, there are two random effects (basin and flood event). The spatial effect is not significant, so this factor can be negligible for the cross-validation procedure. But the flood event is significant and the number of basins affected from a flood event is not balanced, ranging from 1 to 11, and the cross-validation procedure must take this configuration into account. One method to minimize the impact of such limitation is to use the leave-one-out procedure, where the model is fitted with all the observations except the one used for predicting. When the number of basins affected by one flood event is only 1 (in 32 cases) the random effect for this event is not well estimated, implying a worse prediction for these data. We have used the in-sample validation in the main text of the article since the main objective for the development of the climate present model is to use it for prediction. However, we have added the

results of the area under the ROC curve (RA) using the leave-one-out method in the tables and the ROC diagrams for the examples of the two regions in the Appendix section. We have clarified this in the methodology section adding a sentence in the Validation part (line 32, page 8): "The model validation has been carried out using both in-sample and out-of-sample (i.e. leaving-one-out cross-validation) methods."

Reviewer #2. The third question is related to model application in climate scenarios. The results shown correspond to changes in the probability of damaging events. How are those changes computed? The GLM produces the probability of having a "damaging" event, given a precipitation amount P and a basin population R. Therefore, it produces one probability per event. Since the number and nature of events are different in the control and in the future periods, how are the changes in probability computed using the GLM? I would appreciate if the authors could elaborate on this, since the Methodology section ends abruptly after the presentation of precipitation and population projections.

Response: We have used the same relation between variables obtained in the development of the present climate model for the future model. Therefore, we assume that the relation is stationary, however, the values of the variables change, because we incorporate the projections values for precipitation and population. Any other change in the model would involve collecting new data from future observations that we do not have. In the future, when we will have new data, this model can be estimated again by adding new observations to improve it and check its performance. Regarding the random effects, the model used for the future prediction (considering precipitation and population projections data) is the average model, thus, it is based on the fixed effects and does not consider the random effects, which are assigned to zero, their expected value. In order to clarify this point, we have followed the suggestions of the reviewer and add a sentence after the population projections in the Methodology section (line 15, page 11): "After both precipitation and population data for future scenarios have been corrected, the model developed for the present climate has been used to estimate probability of a damaging event in the future. Since the number of flood events and their nature in the future are unknown, the average model has been applied for the prediction."

Reviewer #2. Regarding the climate projections, the authors mention that they selected 30-year periods of EURO-CORDEX simulations starting from the year when the 20-year running mean exceeds the temperature thresholds. These periods are shown on Table 1. However, the SSP population projections are time dependent, but not temperature dependent. Where the analyses made with a different population for each model? Is this a methodological inconsistency? Could the authors provide a brief discussion on this?

Response: The SSP population projections are certainly year dependent, therefore, we have considered the population, regarding to each SSP, of the year of the observation. As each model has its 30-year period window per each level of global warming, we have considered the precipitation observation of these years as well as the population (for the 5 SSP) of the year of each precipitation observation.

Reviewer #2. Regarding data, the authors mention several data sources to identify flood events in the two regions (INUNGAMA, PRESSGAMA and FLOODHYMEX). They seem to use the events identified in these datasets to obtain the damage data provided by the Insurance Compensation Consortium, with a continuous record 1996-2015. Did you check if there are events with relevant damage data in the ICC dataset not included in the other data sources?

Response: As the reviewer mentioned, we have only considered the compensation produced in the basins affected by a flood event that is registered in the flood databases. The CCS database contain more claims than that we have considered, however they are not always are related with a real flood event. For this reason, we have only taken into account the days when we know that a flood event took place.

Reviewer #2. Apart from the above points, there are a few practical details that could

improve the paper: Pag 3, line 14, "summarises"... should be "summarise"?

Response: We have changed it in the manuscript.

Reviewer #2. Pag 13, lines 2-3, "which was affected by 69 flood events between 1996 and 2015, resulting in 171 flood cases". Which is the difference between a "flood event" and a "flood case"?

Response: A flood case is each basin affected by a flood event. Therefore, each observation of the sample. In the page 8 (lines 5-7) of the manuscript is explained in these words: "For each event there can be more than one set of values, depending on the number of affected catchments. From now on we will use the expression "flood case" for each set of values corresponding to a basin affected by a flood event."

Reviewer #2. Pag 16, line 15, "showed". . . should be "shown"?

Response: We have changed it in the manuscript.

Reviewer #2. Pag 22, line 17, "capture". . . should be "captured"?

Response: We have changed it in the manuscript.

Reviewer #2. Pag 26, Basin 130 is missing from the list in Table A2.

Response: We have corrected and added a new row in the table. Thank you very much for your observation.